# DISTRIBUTION-AWARE DIFFUSION MODEL QUANTIZATION VIA DISTORTION MINIMIZATION

## ABSTRACT

Diffusion models have attained significant performance in image/video generation and related tasks. However, while diffusion models excel in delivering excellent results, they suffer from substantial computational complexity due to their large volume of parameters. This poses a significant issue for deployment on mobile devices and hampers the practical applications of diffusion models. In this work, we propose a new post-training quantization approach designed to reduce the computation complexity and memory cost of diffusion models. As the distributions of the outputs of diffusion models differ significantly across timesteps, our approach first splits the timesteps into different groups and optimizes the quantization configuration of each group separately. We then formulate the quantization of each group as a rate-distortion optimization problem to minimize the output distortion caused by quantization given the model size constraint. Because output distortion is highly related to model accuracy, by minimizing the output distortion, our approach is able to compress diffusion models to low bit widths without hurting accuracy. Furthermore, our approach applies Taylor series expansion approximation and proposes an efficient method to find the optimal bit allocation across layers with linear time complexity. Extensive experimentation over four datasets including CIFAR-10, CelebaHQ, LSUN-Bedroom, and LSUN-Church validates the effectiveness of our approach. Empirical results show that our approach obtains a notable improvement over state-of-the-art when the model is quantized to low bit widths.

## 1 INTRODUCTION

Diffusion models ( Song & Ermon (2019); Ho et al. (2020); Song et al. (2020; 2021); Dhariwal & Nichol (2021); Ramesh et al. (2022)) have proven highly effective performance in producing images and videos with both exceptional diversity and fidelity. Recent advancements have showcased their superior results compared to state-of-the-art GAN models, which often struggle with unstable training. As a versatile class of generative models, diffusion models have exhibited their efficacy across a spectrum of applications.

However, the great performance of diffusion models is counterbalanced by their considerable computational consumption. State-of-the-art diffusion models typically demand billions of or even more parameters to obtain a powerful generative neural network. The large volume of parameters, along with its associated memory requirement and computational cost, poses significant challenges for the deployment of diffusion models on mobile devices, especially on resource-constrained platforms such as phones, drones, watches, and self-driving cars. Given the increasing demand for mobile vision applications relying on diffusion models as a cornerstone technique, it is important to compress diffusion models to reduce the time latency and lower the energy consumption.

Quantization ( Han et al. (2015b); Zhu et al. (2017); Zhang et al. (2018); Hubara et al. (2016); Zhou et al. (2017; 2016); Rastegari et al. (2016)) is one of the powerful methods for model compression. Instead of utilizing the original 32-bit floating-point representation, parameters can be quantized into lower bit-widths such as 8-bit or even less to reduce model size and computation. More importantly, Post-Training Quantization (PTQ) eliminates the need of re-training, which makes quantization more practical and can significantly save the cost, as re-training itself is very expensive and even not possible in scenarios where retraining data is not available. Although a lot of post-training

quantization methods have been proposed, most of them are developed for Convolutional Neural Networks (CNNs) and Vision Transformers (ViTs). Given that diffusion models possess a distinct network architecture with unique features, employing Post-Training Quantization methods designed for CNNs and ViTs is not optimal, as they fail to capitalize on the nature of diffusion models. One main feature of diffusion models is that diffusion models have many timesteps and the distributions of layer outputs differ significantly across timesteps.

Several works studying post-training quantization methods tailored for diffusion models have been propossed, such as PTQ4DM Shang et al. (2023), Q-Diffusion Li et al. (2023a), PTQD He et al. (2023), Q-DM Li et al. (2023b), APQ-DM Wang et al. (2024), and TFMQ-DM Huang et al. (2024). However, one common issue in prior works is that they use equal bit width to quantize weights and activations of all layers. Because parameters in different layers react very distinctively to quantization, it is more reasonable to use mixed precision for quantization. Moreover, in prior works, the output distortion caused by quantization is not minimized. As output distortion is highly related to accuracy, large distortion may lead to serious accuracy loss.

In this paper, we propose a novel post-training quantization approach for diffusion models. Our approach adopts mixed-precision quantization and uses different bit widths to quantize parameters in different layers. As the distributions of activations differ significantly across timesteps, we splits the timesteps into groups and optimizes the quantization configuration of each step separately. We formulate the quantization of each timestep as a rate-distortion optimization problem to maximally maintain the accuracy when quantized to low bit widths by minimizing the output distortion caused by quantization. To solve the optimization problem efficiently, our approach applys first-order Taylor approximation where an important additivity property is observed, that is the output distortion caused by quantizing all layers equals the sum of output distortion due to the quantization of each single layer. Utilizing such additivity property, we propose an efficient optimization method with only linear time complexity to find the solution.

Extensive experiments have been conducted on four datasets to validate the effectiveness of our approach. Our approach advances the state-of-the-art noticeably and can reduce the bit width to 6 bits on diffusion models without compromising accuracy. The key contributions of this paper are summarized as follows:

- We propose a novel post-training quantization approach for diffusion model compression. First, our approach adopts mixed precision to quantize parameters in different layers. We then split the timesteps into groups and optimize the quantization configuration of each timestep separately. The quantization of each timestep is formulated as a rate-distortion optimization problem, where output distortion caused by quantization is minimized. To the best of our knowledge, this is the first work which explicitly proposes rate-distortion optimization for the quantization of diffusion models.

- We propose a very efficient method to solve the optimization problem, where we apply first-order Taylor approximation and observe an important additivity property. A mathematical derivation is provided for the additivity property. By utilizing the additivity property, we develop an efficient algorithm with only linear time complexity to find the solution.

- Extensive experiments have been conducted to demonstrate the effectiveness. Our approach noticeably outperforms state-of-the-art over four benchmark datasets.

## 2 RELATED WORKS

In this section, we discuss prior works relevant to our paper. We first review post-training quantization methods designed for Convolutional Neural Networks, where most of post-training quantization works lie in this category. We then discuss the post-training quantization works developed for diffusion models. Our approach belongs to the second category.

### 2.1 POST-TRAINING QUANTIZATION FOR CONVOLUTIONAL NEURAL NETWORKS

Most of post-training quantization works belong to this category. Choukroun et al. devised an approach that treats quantization as a Minimum Mean Squared Error (MMSE) problem Choukroun et al. (2019), allowing for low-bit precision inference without requiring retraining. Banner et al.

introduced the first practical 4-bit post-training quantization method Banner et al. (2019) through a combination of analytical clipping, per-channel bit allocation, and bias correction. Zhao et al. improved post-training quantization by introducing Outlier Channel Splitting (OCS) Zhao et al. (2019), a technique that duplicates channels containing outliers and reduces their values by half. Nagel et al. introduced adaptive rounding as a refined weight rounding mechanism for post-training quantization Nagel et al. (2020), dynamically adjusting to both data and task loss for enhanced effectiveness. Additionally, Nagel et al. proposed data-free quantization Nagel et al. (2019) by incorporating weight equalization and bias correction.

Several studies have also been conducted to achieve extremely low bit widths through post-training quantization. Cai et al. (2020) introduced Zero-Shot Quantization (ZSQ) as a method for achieving ultra-low-bit neural networks by distilling datasets based on statistics derived from batch normalization layers. Fang et al. (2020) employed piecewise uniform quantization for weights, resulting in improved accuracy on ImageNet using a 2-piece configuration. Li et al. (2023a) leveraged the Hessian of cross-layer weights to establish block-wise layer dependencies when optimizing the quantizer. Zhong et al. (2022) proposed an innovative fine-tuning loss that preserved multiclass discriminative qualities through the integration of synthetically generated data. Oh et al. (2022) utilized a logarithmic quantization scheme, complemented by corresponding shifter arithmetic operations, to replace multipliers, thereby enabling higher arithmetic precision while maintaining low representative precision. Lin et al. (2023) introduced a bit-shrinking algorithm aimed at gradually quantizing networks to lower bit-widths, which helps flatten the loss landscape and mitigates the risk of encountering suboptimal local minima during the fine-tuning process. Ma et al. (2023) employed orthogonal linear optimization to efficiently search bit-allocation in mixed-precision post-training quantization, resulting in significant time savings.

The above approaches mainly targeted the quantization of Convolutional Neural Networks. However, directly applying them to quantize diffusion models may yield sub-optimal outcomes. This is because Diffusion Models have a distinct network architecture with unique features, which is different with Convolutional Neural Networks.

## 2.2 Post-training Quantization for Diffusion Models

Several Post-training Quantization methods have been proposed recently for Diffusion Models. Li et al. (2023a) proposed Q-Diffusion, which is a PTQ method tailored specifically to the distinctive multi-timestep pipeline and the architecture of diffusion models. Their approach aims to compress the noise estimation network to expedite the generation process. They identified the primary challenge of quantizing diffusion models with dynamic output distributions across multiple time steps and the bimodal activation distribution of shortcut layers. To tackle these challenges, they employed timestep-aware calibration and split shortcut quantization.

On the other hand, Shang et al. departed from traditional training-aware compression methods and introduced a post-training quantization approach named PTQ4DM Shang et al. (2023) to accelerate diffusion models. They focused on developing a diffusion-model-specific PTQ method by examining PTQ applied to diffusion models from three perspectives: quantized operations, calibration dataset, and calibration metric. Drawing insights from their comprehensive investigations, they synthesized and utilized several strategies to devise their method, particularly targeting the unique multi-time-step structure of diffusion models. He et al. (2023) recognized that at each denoising step, quantization noise causes deviations in the estimated mean and mismatches with the predetermined variance schedule. To address this, PTQD is proposed to eliminate quantization noise through the use of correlated noise and residual noise correction techniques. Li et al. (2023b) identified that the bottlenecks in low-bit quantized diffusion models stem from significant distribution oscillations in activations and quantization errors accumulate during the multi-step denoising process. To address these issues, they propose Q-DM, an efficient low-bit quantized diffusion model that achieves a high compression ratio while maintaining competitive performance in image generation tasks.

To address key limitations in conventional quantization frameworks by designing distribution-aware quantization functions and optimizing calibration timesteps, Wang et al. (2024) propose an accurate post-training quantization framework for diffusion models (APQ-DM). Significant improvement of image generation performance has been achieved with minimal computational overhead. To mitigate the inefficiencies of PTQ in diffusion models, which typically suffer from extended inference

times and high memory demands, Huang et al. (2024) proposed Temporal Feature Maintenance Quantization (TFMQ-DM). This method enhanced compression efficiency and preserved temporal information by focusing on time-step-specific features, achieving near full-precision model performance under 4-bit weight quantization. Zhao et al. (2024) introduced MixDQ, a mixed-precision quantization method that handles both the imbalance sensitivity and alignment degradation problems for diffusion quantization. Sui et al. (2024) developed a novel weight quantization method that quantizes the UNet from Stable Diffusion v1.5 to 1.99 bits.

# 3 Approach

In this section, we present our post-training quantization approach designed for diffusion models. We first introduce the formulation of quantization in our approach where quantization is formulated as a rate-distortion optimization problem. We then present the algorithm to solve the optimization problem and discuss the time complexity of our method. As the distributions of activation are different across the time steps, we thus divide the timesteps into groups and optimize the quantization of each group separately. Then for each of the groups, we apply mixed precision and find the optimal bit widths for the layers, given the fact that different layers react differently to quantization.

## 3.1 Preliminaries

**Diffusion Models** apply a Markov chain to generate images and videos where the forward diffusion process adds Gaussian noise to data $\mathbf{x}_0 \sim q(\mathbf{x}_0)$ for $T$ times, resulting in noisy samples $\mathbf{x}_0, ..., \mathbf{x}_T$:

$$q(\mathbf{x}_t|\mathbf{x}_{t-1}) = \mathcal{N}(\mathbf{x}_t; \sqrt{1 - \beta_t}\mathbf{x}_{t-1}, \beta t\mathbf{I}) \tag{1}$$

in which $\beta_t \in (0, 1)$ denotes the variance schedule handling the strength of the Gaussian noise of each step. On the other hand, diffusion models have a reverse process to remove the noise from a sample from the Gaussian noise input $\mathbf{x}_T \in (\mathbf{0}, \mathbf{1})$ to generate images gradually. However, the reverse conditional distribution $q(\mathbf{x}_{t-1}|\mathbf{x}_t)$ is unavailable, as a result, diffusion models sample from a learned conditional distribution $p_\theta(\mathbf{x}_{t-1}|\mathbf{x}_t) = \mathcal{N}(\mathbf{x}_{t-1}; \tilde{\mathbf{u}}_{\theta,t}(\mathbf{x}_t), \tilde{\beta}_t\mathbf{I})$.

**Quantization** aims to use smaller bit widths to represent each parameter. Instead of using 32-bit or 16-bit floating point representation, one can quantize parameters into lower bit widths (say 8 bits or 4 bits) to compress model size and reduce computation. In this work, we quantizes all the weights and inputs (activations) involved in each layer. Following prior quantization scheme, we do not quantize biases in normalization layer, as the volume of the parameters contained in these layers is negligible. More specifically, our approach adopts uniform scalar quantization to quantize parameters. Given input $\mathbf{W}$, the quantized value is defined as,

$$q(\mathbf{W}) = \Delta \cdot Clip(\lfloor \frac{\mathbf{W}}{\Delta} \rfloor, -2^{b-1}, 2^{b-1} - 1), \tag{2}$$

where $\Delta$ is the quantization step size and $b$ is the bit width. $\Delta$ and $b$ are two hyper-parameters. Clip denotes the clipping function which clips the elements that exceed ranges.

## 3.2 Formulation with Distortion Minimization

Our approach first splits timesteps into groups and optimizes the quantization of each group separately. Specifically, we adopt mixed precision to quantize parameters in different layers. The quantization of each timestep group is formulated as a rate-distortion optimization problem where the learning objective is to minimize output distortion caused by quantization. Let $\mathbf{O}$ denote the output of the original model and $\widehat{\mathbf{O}}$ denote the output of that quantized model, our approach aims to minimize the output distortion when parameters are quantized. The output distortion is defined as the Mean Square Error (MSE) added by the Structural Similarity Index Measure (SSIM) loss between $\mathbf{O}$ and $\widehat{\mathbf{O}}$,

$$\Gamma(\mathbf{O}, \widehat{\mathbf{O}}) = \frac{\|\mathbf{O} - \widehat{\mathbf{O}}\|^2}{\|\mathbf{O}\|_F^2 \cdot \|\widehat{\mathbf{O}}\|_F^2} + \frac{(2u_{\mathbf{O}}u_{\widehat{\mathbf{O}}} + c_1)(2\sigma_{\mathbf{O}\widehat{\mathbf{O}}} + c_2)}{(u_{\mathbf{O}}^2 + u_{\widehat{\mathbf{O}}}^2 + c_1)(\sigma_{\mathbf{O}}^2 + \sigma_{\widehat{\mathbf{O}}}^2 + c_2)} \tag{3}$$

where $u_{\mathbf{O}}$ is the pixel sample mean of $\mathbf{O}$, $u_{\widehat{\mathbf{O}}}$ is the pixel sample mean of $\widehat{\mathbf{O}}$, $\sigma_{\mathbf{O}}$ is the variance of $\mathbf{O}$, $\sigma_{\widehat{\mathbf{O}}}$ is the variance of $\widehat{\mathbf{O}}$, $\sigma_{\mathbf{O}\widehat{\mathbf{O}}}$ is the covariance of $\mathbf{O}$ and $\widehat{\mathbf{O}}$, and $c_1$ and $c_2$ are two variables to

stabilize the division. By directly optimizing the output distortion between $\mathbf{O}$ and $\widehat{\mathbf{O}}$, our approach can well maintain the accuracy when models are quantized to low bit widths.

Furthermore, we adopt mixed precision to quantize parameters in different layers. Because parameters in different layer react very distinctively to quantization, for layers which are sensitive to quantize, large bit widths should be allocated to quantize them. While for other layers which are less sensitive, one can allocate small bit widths to quantize. As a result, using equal bit width to quantize parameters in all layers is not reasonable. The quantization with mixed precision is formulated as a rate-distortion optimization problem where output distortion is minimized, given the size constraint,

$$\underset{q(\mathbf{W}_1^\dagger),...,q(\mathbf{W}_l^\dagger),q(\mathbf{I}_1^\dagger),...,q(\mathbf{I}_l^\dagger)}{\arg\min} \Gamma_{\mathbf{W}_1^\dagger,...,\mathbf{W}_l^\dagger,\mathbf{I}_1^\dagger,...,\mathbf{I}_l^\dagger}(\mathbf{O},\widehat{\mathbf{O}})$$

$$s.t. \sum_{i=1}^l s_{\mathbf{W}_i^\dagger} + \sum_{i=1}^l s_{\mathbf{I}_i^\dagger} \leq S, \tag{4}$$

where $S$ denotes the total size of the model after quantization, $s_{\mathbf{W}_i^\dagger}$ is the size of quantized weights $\mathbf{W}_i^\dagger$ (i.e., $q(\mathbf{W}_i^\dagger)$) of layer $i$ which equal to the number of weights multiplied by the bit width of this layer, $s_{\mathbf{I}_i^\dagger}$ is the size of quantized activations $\mathbf{I}_i^\dagger$ (i.e., $q(\mathbf{I}_i^\dagger)$) of layer $i$ which equal to the number of activations multiplied by the bit width of this layer, and $l$ is the total number of layers.

### 3.3 FIRST ORDER TAYLOR APPROXIMATION

Directly optimizing 4 is difficult because of the huge search space of bit widths, which increases exponentially as the number of layers where in practice it can be tens of thousand or even more. We utilize first-order Taylor approximation and observe an important additivity property. As a result, the output distortion $\Gamma_{\mathbf{W}_1^\dagger,...,\mathbf{W}_l^\dagger,\mathbf{I}_1^\dagger,...,\mathbf{I}_l^\dagger}(\mathbf{O},\widehat{\mathbf{O}})$ can be rewritten in the following way: Actually, by utilizing Taylor Series Expansion approximation, output distortion $\Gamma_{\mathbf{W}_1^\dagger,...,\mathbf{W}_l^\dagger,\mathbf{I}_1^\dagger,...,\mathbf{I}_l^\dagger}(\mathbf{O},\widehat{\mathbf{O}})$, caused by quantizing all filters and regions, can be decomposed to the sum of output distortions due to the quantization of each individual item

**Property 1.** *The output distortion caused by quantizing all layers equal to the sum of output distortions due to the quantization of each individual layer,*

$$\Gamma_{\mathbf{W}_1^\dagger,...,\mathbf{W}_l^\dagger,\mathbf{I}_1^\dagger,...,\mathbf{I}_l^\dagger}(\mathbf{O}_l,\widehat{\mathbf{O}}_l) = \sum_{i=1}^l \Gamma_{\mathbf{W}_i^\dagger}(\mathbf{O},\widehat{\mathbf{O}}) + \sum_{i=1}^l \Gamma_{\mathbf{I}_i^\dagger}(\mathbf{O},\widehat{\mathbf{O}}) \tag{5}$$

*if the model is continuously differentiable in every layer and quantization errors can be considered as small deviations distributed with zero mean.*

**Mathematical Derivation**. The proof of 5 is as follows. Let $\mathcal{F}(\mathbf{W}_1,\mathbf{W}_2,...,\mathbf{W}_l)$ denote a diffusion model weighted by $l$ layers and $\widetilde{\mathcal{F}}(\mathbf{W}_1,\mathbf{W}_2,...,\mathbf{W}_l,s_1,s_2,...,s_l)$ denote a modified diffusion model of $\mathcal{F}$ where an element-wise add layer with parameter $s_i$ is followed for each input region $a_i$. Based on this definition, we have

$$\mathcal{F}(\mathbf{W}_1,\mathbf{W}_2,...,\mathbf{W}_l) = \widetilde{\mathcal{F}}(\mathbf{W}_1,\mathbf{W}_2,...,\mathbf{W}_l,0,0,...,0) \tag{6}$$

Define two variables $X_0$ and $\Delta X$, where $X_0 = (W_1,...,W_l,0,...,0)$ and $\Delta X = (\Delta W_1,...,\Delta W_l,\Delta s_1,...,\Delta s_l)$. Here we use $\Delta W_i$ and $\Delta s_i$ denote the quantization errors of weights and activations, respectively. Assume that the quantization error can be considered as small deviation. We apply the Taylor series expansion up to first order term on $\widetilde{\mathcal{F}}$ at $X_0$,

$$\widetilde{\mathcal{F}}(X_0+\Delta X) - \widetilde{\mathcal{F}}(X_0) = \sum_i \frac{\partial \widetilde{\mathcal{F}}}{\partial W_i} \cdot \Delta W_i + \sum_i \frac{\partial \widetilde{\mathcal{F}}}{\partial s_i} \cdot \Delta s_i \tag{7}$$

Then $\|\widetilde{\mathcal{F}}(X_0+\Delta X) - \widetilde{\mathcal{F}}(X_0)\|^2$ can be written as

$$\left(\sum_i \Delta W_i^\top \cdot \frac{\partial \widetilde{\mathcal{F}}}{\partial W_i}^\top + \sum_i \Delta s_i^\top \cdot \frac{\partial \widetilde{\mathcal{F}}}{\partial s_i}^\top\right) \cdot \left(\sum_i \frac{\partial \widetilde{\mathcal{F}}}{\partial W_i} \cdot \Delta W_i + \sum_i \frac{\partial \widetilde{\mathcal{F}}}{\partial s_i} \cdot \Delta s_i\right) \tag{8}$$

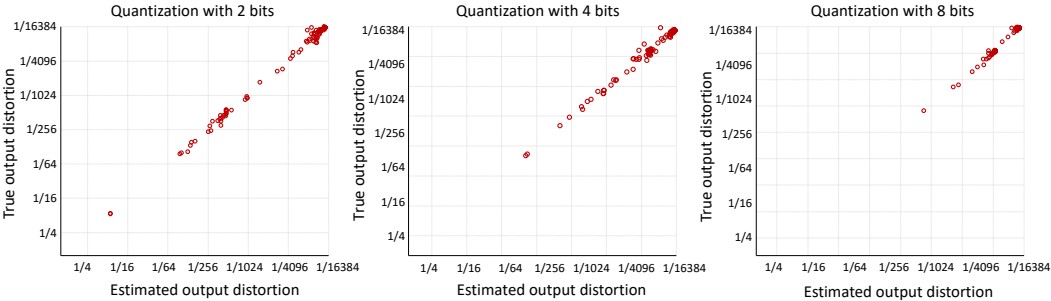

Figure 1: Demonstration of the additivity property of output distortion. The vertical axis represents the left side of Equation (5). The horizontal axis represents the right side of Equation (5).

Because quantization errors in different layers are independently distributed with zero mean, the cross terms of (8) disappear when taking the expectation. That is:

$$E(\Delta W_i^\top \cdot \frac{\partial \widetilde{\mathcal{F}}}{\partial W_i}^\top \cdot \frac{\partial \widetilde{\mathcal{F}}}{\partial W_j} \cdot \Delta W_j) = E(\Delta W_i^\top) \cdot \frac{\partial \widetilde{\mathcal{F}}}{\partial W_i}^\top \cdot \frac{\partial \widetilde{\mathcal{F}}}{\partial W_j} \cdot E(\Delta W_j) = 0 \tag{9}$$

as is the case also for the cross products between $W_i$ and $s_j$ (all $i$, $j$), and $s_i$ and $s_j$ ($i \neq j$). Then, we can obtain

$$E(\|\widetilde{\mathcal{F}}(X_0 + \Delta X) - \widetilde{\mathcal{F}}(X_0)\|^2) = \sum_i E\Big(\|\frac{\partial \widetilde{\mathcal{F}}}{\partial W_i} \cdot \Delta W_i\|^2\Big) + \sum_i E\Big(\|\frac{\partial \widetilde{\mathcal{F}}}{\partial s_i} \cdot \Delta s_i\|^2\Big) \tag{10}$$

Eq. (10) is the result we want because, again, according to the Taylor series expansion up to first order terms, we have

$$\frac{\partial \widetilde{\mathcal{F}}}{\partial W_i} \cdot \Delta W_i = \widetilde{\mathcal{F}}(..., W_i + \Delta W_i, ..., W_l, 0, ...) - \widetilde{\mathcal{F}}(..., W_i, ..., W_l, 0, ...) \tag{11}$$

Similarly, we have another equation for inputs,

$$\frac{\partial \widetilde{\mathcal{F}}}{\partial s_i} \cdot \Delta s_i = \widetilde{\mathcal{F}}(W_1, ..., W_l, 0, ..., \Delta s_i, ...) - \widetilde{\mathcal{F}}(W_1, ..., W_l, 0, ..., 0, ...) \tag{12}$$

After dividing both sides of (10) by the dimensionality of the output vector of the neural network, the left side becomes $\Gamma_{\mathbf{W}_1^\dagger, ..., \mathbf{W}_n^\dagger, \mathbf{I}_1^\dagger, ..., \mathbf{I}_m^\dagger}(\mathbf{O}, \widehat{\mathbf{O}})$ and the right side becomes the sum of all output distortion due to the quantization of each individual filters and regions. Figure 1 shows an illustration of the additivity property. In Figure 1, all the points are closed to the diagonal, meaning that the additivity property holds.

### 3.4 OPTIMIZATION AND COMPLEXITY ANALYSIS

After utilizing the additivity property in 5, the formulation can be rewritten as

$$\underset{q(\mathbf{W}_1^\dagger), ..., q(\mathbf{W}_l^\dagger), q(\mathbf{I}_1^\dagger), ..., q(\mathbf{I}_l^\dagger)}{\arg\min} \sum_{i=1}^{l} \Gamma_{\mathbf{W}_i^\dagger}(\mathbf{O}, \widehat{\mathbf{O}}) + \sum_{i=1}^{l} \Gamma_{\mathbf{I}_i^\dagger}(\mathbf{O}, \widehat{\mathbf{O}})$$

$$s.t. \sum_{i=1}^{l} s_{\mathbf{W}_i^\dagger} + \sum_{i=1}^{l} s_{\mathbf{I}_i^\dagger} \leq S, \tag{13}$$

We adopt Lagrangian formulation to solve (13). The Lagrangian cost function of (13) is defined as

$$\sum_{i=1}^{l} \Gamma_{\mathbf{W}_i^\dagger}(\mathbf{O}, \widehat{\mathbf{O}}) + \sum_{i=1}^{l} \Gamma_{\mathbf{I}_i^\dagger}(\mathbf{O}, \widehat{\mathbf{O}}) - \gamma \cdot (\sum_{i=1}^{l} s_{\mathbf{W}_i^\dagger} + \sum_{i=1}^{l} s_{\mathbf{I}_i^\dagger}), \tag{14}$$

where $\gamma$ decides the trade-off between the output error and the total size of the network. We then set the partial derivatives of (14) to zero w.r.t. each $s_{\mathbf{W}_i^\dagger}$ and $s_{\mathbf{I}_i^\dagger}$ to find the extreme value. We then have the optimal condition,

$$\frac{\partial \Gamma_{\mathbf{W}_i^\dagger}(\mathbf{O}, \widehat{\mathbf{O}})}{\partial s_{\mathbf{W}_i^\dagger}} = \frac{\partial \Gamma_{\mathbf{I}_i^\dagger}(\mathbf{O}, \widehat{\mathbf{O}})}{\partial s_{\mathbf{I}_i^\dagger}} = \gamma. \tag{15}$$

The above equation expresses the extreme value condition, which is that the slopes of output error versus size curves must be equal. Based on this condition, we can find the solution by generating the output error versus the size curve for each filter and region and choosing the points with an equal slope on the curves.

According to 15, we solve the joint optimization problem by enumerating $\gamma$ and selecting the point with slope equal to $\gamma$ on each output error versus size curve. We generate the output error versus size curves for all layers' weights and activations. An output error versus size curve is a discrete curve with $B$ points, by setting the quantization bit width from 1-bit to $B$-bits. We enumerate different $\gamma$ to find the best one with minimal output error and without exceeding the constraint of model size. Let $N$ denotes the number of curves and $M$ denotes the number of points in each curve. The time complexity to solve the joint optimization problem is $O(K \cdot M \cdot N)$, where $K$ is the total number of slope $\gamma$ to be evaluated. This indicates that our solution is linear time complexity and independent on the number of parameters.

## 4 EXPERIMENTS

**Datasets and models.** We perform image generation experiments using two widely-adopted diffusion models: Denoising Diffusion Probabilistic Models (DDPM) Ho et al. (2020) and Denoising Diffusion Implicit Models (DDIM) Song et al. (2021). Both models train the image denoiser over the course of 1,000 time steps. To evaluate the performance of these models, we utilize four well-established benchmark datasets. The first is CIFAR-10 (32×32) Krizhevsky (2009), which contains 50,000 training images across 10 classes. Next, we use the CelebA-HQ dataset Liu et al. (2015) at a resolution of 256×256, consisting of 30,000 high-quality celebrity face images. Additionally, we incorporate LSUN-Bedrooms and LSUN-Churches Fisher et al. (2015), both with image resolutions of 256×256, each providing 50,000 training images. These datasets enable a comprehensive evaluation of model performance across diverse image types, from natural scenes to human faces.

**Quantization settings.** Following the approach of previous work Huang et al. (2024), we apply channel-wise quantization to the model's weights and layer-wise quantization to its activations. These quantization strategies help reduce the model's memory footprint and computational requirements while maintaining performance. Additionally, based on empirical findings from traditional model quantization techniques Han et al. (2015a); Mohammad et al. (2016), we preserve the input and output layers in full precision (FP) to prevent significant degradation in accuracy. Maintaining these critical layers in FP ensures that the model retains high-quality data representation at the beginning and end of the network, where precision is most impactful.

**Evaluation metrics.** For each experiment, we assess the performance of the diffusion models using the Frechet Inception Distance (FID), a widely recognized metric for evaluating the quality of images generated by generative models. FID measures the distance between the distribution of generated images and that of real images (referred to as the 'ground truth'), providing insight into both the fidelity and diversity of the generated samples. Lower FID scores indicate closer alignment between the generated and real images, signifying higher image quality.

In addition to FID, we also report the Inception Score (IS) as a complementary metric. Unlike FID, IS evaluates the generated images based solely on their own distribution, without comparison to real images. It assesses both the recognizability of the generated images and the diversity within the set, making it a useful reference for understanding model performance in isolation. By using both FID and IS, we provide a comprehensive evaluation of the generative capabilities. All experiments are conducted using a single NVIDIA-A100-SXM4 GPU and implemented in the PyTorch framework.

Table 1: Experiment on 4/5/6/7/8-bit quantized diffusion models generating image on four datasets. "Bits" denotes the bit-width of weights/activations.

| Model | Dataset | Bits | FID↓ | IS↑ | Model | Dataset | Bits | FID↓ | IS↑ |
|---|---|---|---|---|---|---|---|---|---|
| DDPM | CIFAR-10 | FP | 3.30 | 9.47 | DDIM | LSUN-Bedroom | FP | 7.89 | 2.81 |
| | | 8/8 | **3.16** | 9.48 | | | 8/8 | 8.09 | 2.85 |
| | | 7/7 | **3.17** | 9.43 | | | 7/7 | 8.21 | 2.78 |
| | | 6/6 | 3.37 | 9.29 | | | 6/6 | 9.37 | 2.32 |
| | | 5/5 | 6.13 | 9.01 | | | 5/5 | 14.65 | 1.83 |
| | | 4/4 | 33.03 | 6.76 | | | 4/4 | 39.88 | 1.51 |
| | Celeba-HQ | FP | 9.01 | 2.50 | | LSUN-Church | FP | 11.33 | 2.74 |
| | | 8/8 | **8.89** | 2.58 | | | 8/8 | 11.56 | 3.05 |
| | | 7/7 | 9.02 | 2.56 | | | 7/7 | 13.03 | 2.83 |
| | | 6/6 | 9.06 | 2.54 | | | 6/6 | 15.11 | 2.24 |
| | | 5/5 | 9.30 | 2.55 | | | 5/5 | 24.65 | 1.67 |
| | | 4/4 | 9.49 | 2.81 | | | 4/4 | 59.74 | 1.08 |

## 4.1 ABLATION STUDY

Table 1 presents the quantitative results of the proposed method on four used datasets. As shown, the low-bit quantized models (i.e., models using 5 or fewer bits) for generalization, such as DDPM on CIFAR-10 and DDIM on LSUN-Bedrooms and LSUN-Churches exhibit a significant performance degradation in the image generation task compared to their full precision (FP) counterparts.

For instance, on CIFAR-10, there is a performance gap of 2.83 and 29.73 in FID score when using 5-bit and 4-bit quantization, respectively. Surprisingly, on CelebA-HQ, our proposed quantization method allows DDPM to generate images comparable to the FP model, even with 5-bit and 4-bit quantization (0.29 and 0.48 performance cap in terms of FID score respectively). On the LSUN-Bedroom and LSUN-Church datasets, the performance degradation becomes pronounced when the bit precision is reduced to 6 or lower. Specifically, the FID scores show a significant decline, indicating that the quality of generated images deteriorates considerably with lower bit quantization.

This performance drop is likely due to the reduced capacity of low-bit models to capture fine-grained details in the data, leading to a loss in image quality and diversity. While quantization effectively reduces model size and computational cost, the results indicate that extreme low-bit quantization introduces challenges in maintaining the fidelity of generated images.

## 4.2 MAIN RESULTS

The experimental results are presented in Table 2. We compare our method against baseline approaches within the same frameworks, such as DDPM on CIFAR-10 and CelebA-HQ, and DDIM on LSUN-Bedroom and LSUN-Church, for the image generation task. The baseline methods include Q-Diffusion Li et al. (2023a), PTQ4DM Shang et al. (2023), PTQD He et al. (2023) and APQ-DM Wang et al. (2024).

From the table, we can observe the following results for W8A8 quantization: (1) On CIFAR-10, compared to the full precision (FP) model, our method achieves a 0.14 reduction in FID while maintaining the same IS. When compared to the state-of-the-art (SOTA) method, PTQ4DM, our approach yields a 0.12 FID improvement. (2) On CelebA-HQ, our method outperforms FP with a 0.12 reduction in FID and a 0.08 increase in IS, while also surpassing the SOTA (PTQ4DM) by 0.05 in FID. (3) On LSUN-Bedroom, our method shows a slight 0.2 increase in FID compared to FP, yet it achieves the closest FID score among all SOTA methods. (4) On LSUN-Church, APQ-DM achieves the best performance, surpassing the FP in terms of FID, while our method performs comparably to PTQD. We can also observe that the performance trends across all methods for W6A6 are similar to those observed for W8A8.

## 4.3 VISUALIZATION

Figure 2 demonstrates some examples of the images that are generated by different quantized Diffusion models (DDPM and DDIM), where our method can still acquire plausible images with high-quality details with weights and activations in low bitwidths (8bits and 6bits). For example, on

Table 2: Comparisons with the state-of-the-arts data-free post-training quantization methods on image generation for DDPM and DDIM diffusion models across various datasets and bitwidth setting. It is important to note that the FID and IS scores for all methods were obtained by applying them to the same baseline model.

| Method | Bitwidth | CIFAR-10 | | CelebA-HQ | | LSUN-Bedroom | | LSUN-Church | |
|---|---|---|---|---|---|---|---|---|---|
| | | FID↓ | IS↑ | FID↓ | IS↑ | FID↓ | IS↑ | FID↓ | IS↑ |
| Baseline | FP | 3.3 | 9.47 | 9.01 | 2.50 | 7.89 | 2.81 | 11.33 | 2.74 |
| Q-Diffusion | | 3.72 | 9.26 | 9.16 | 2.31 | 8.69 | 2.60 | 12.48 | 2.70 |
| PTQ4DM | | 3.28 | 9.21 | 8.94 | 2.33 | 9.23 | 2.56 | 13.20 | 2.74 |
| PTQD | W8A8 | 3.89 | 9.27 | 9.81 | 2.25 | 9.86 | 2.41 | 11.51 | 2.61 |
| APQ-DM | | 3.31 | 9.47 | 9.33 | 2.47 | 8.92 | **2.94** | **11.16** | 2.72 |
| ours | | **3.16** | **9.48** | **8.89** | **2.58** | **8.09** | 2.85 | 11.56 | **3.05** |
| Q-Diffusion | | 5.16 | 8.96 | 13.81 | 2.28 | 11.04 | 2.09 | 19.59 | 2.52 |
| PTQ4DM | | 6.04 | 8.92 | 14.75 | 2.24 | 10.59 | 2.11 | 19.05 | 2.48 |
| PTQD | W6A6 | - | - | - | - | - | - | - | - |
| APQ-DM | | 4.12 | **9.46** | 9.97 | 2.36 | 9.88 | 2.27 | **14.36** | **2.65** |
| ours | | **3.37** | 9.29 | **9.06** | **2.54** | **9.37** | **2.32** | 15.11 | 2.24 |

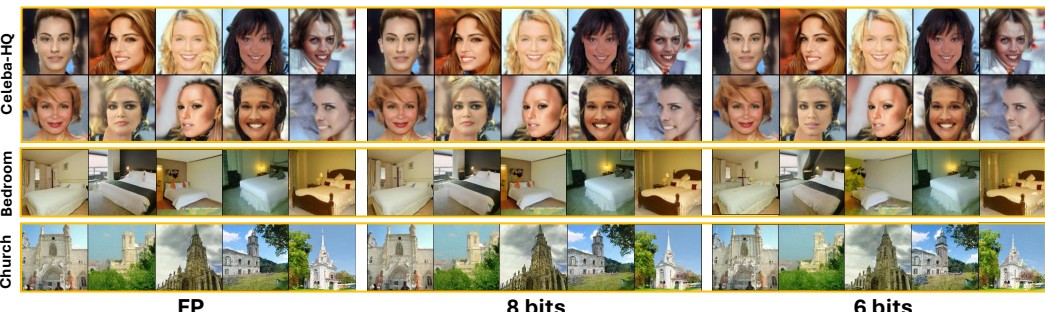

Figure 2: Random samples from W8A8 and W6A6 quantized and full-precision DDPM on CelebA-HQ and DDIM on LSUN-Bedroom and Church.

CelebA-HQ, the images generated by our method with 6-bit quantization maintain nearly the same fidelity as those produced by the FP model.

On LSUN-Bedroom and LSUN-Church, the images generated by our method with 8-bit quantization maintain the same quality as those from the FP model. However, with 6-bit quantization, the generated images experience a slight degradation in quality. Additional visualization results are provided in the Appendix.

## 5  CONCLUSIONS

While diffusion models achieve impressive performance in image and video generation, their practical application is hindered by high computational complexity and memory demands. To overcome these limitations, we propose a post-training quantization approach that reduces both computational and memory requirements. Our method groups timesteps and optimizes their quantization configurations independently, effectively minimizing output distortion while preserving accuracy at lower bit widths.

We formulate the quantization of each group as a rate-distortion optimization problem and develop an efficient algorithm to find the solution by apply first order Taylor approximation. Experiments on various datasets show that our approach reduces the bit width to 5-6 bits while maintaining high accuracy, outperforming state-of-the-art methods.

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

# A  APPENDIX

## A.1  DISTORTION CURVES OF ACTIVATION ACROSS LAYERS WITH VARIOUS TIMESTEPS

As mentioned earlier, we have observed that the activation distribution varies significantly across different timesteps. In Figure 3, we illustrate the distribution of activation quantization distortion across layers, measured at intervals of 100 timesteps from 100 to 1000 on the CIFAR-10 dataset. This example highlights the variability in distortion at a fixed bit-width (i.e., 2 bits). Notably, the level of distortion fluctuates across layers depending on the timestep, indicating that a uniform quantization approach fails to capture the temporal nuances in activation distribution. This reinforces the need for timestep-specific quantization strategies to mitigate distortion and maintain performance consistency throughout the denoising process in diffusion models.

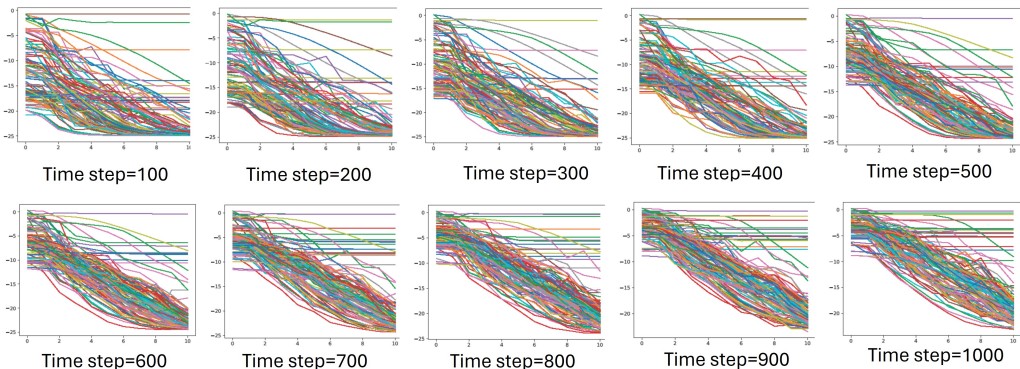

Figure 3: **Distribution of activation quantization distortion across layers using our method on the CIFAR-10 dataset at various time steps during the denoising process.**

## A.2  MORE VISUALIZATION RESULTS

In Figures 4 to 8, we present additional visualization results on CIFAR-10, Celeba-HQ, LSUN-Bedroom, LSUN-Church and ImageNet.

For the CIFAR-10 dataset, we display image generation results using the DDPM model with 8, 7, 6, 5, and 4 bits, applying our quantization method. We can see that, compared to the results generated from the FP model, the images produced at 6 bits are acceptable. However, once the bit-width drops to 5 bits, the quality of the image generation declines significantly.

On the CelebA-HQ dataset, we observe that even at a bit-width of 5, the quality of image generation remains acceptable when compared to images generated by the FP model. Despite the reduced bit-width, the visual fidelity is well-preserved, demonstrating the robustness of the quantization method in maintaining image quality under lower precision settings. This highlights the effectiveness of the approach in balancing compression and performance. As the bit-width continues to drop to 4, the quality of the generated images degrades significantly. The visual artifacts become more prominent, and the fine details present in higher bit-width settings are lost. This substantial decline in image quality underscores the challenges of maintaining performance at extremely low precision levels, highlighting the trade-off between compression efficiency and visual fidelity.

On the LSUN-Bedroom dataset, it is evident that the fidelity of images generated by the DDIM model with 8/7/6-bits quantization remains acceptable, maintaining a reasonable level of detail and visual quality. However, when the bit-width is reduced to 5 bits, the degradation in image quality becomes noticeable. Some of the generated images, such as the second and fifth examples, exhibit significant artifacts and distortions, rendering them unacceptable. This demonstrates the sensitivity of the model to lower bit-widths, where even a slight reduction in precision can lead to a marked drop in generation performance, especially in more complex scenes.

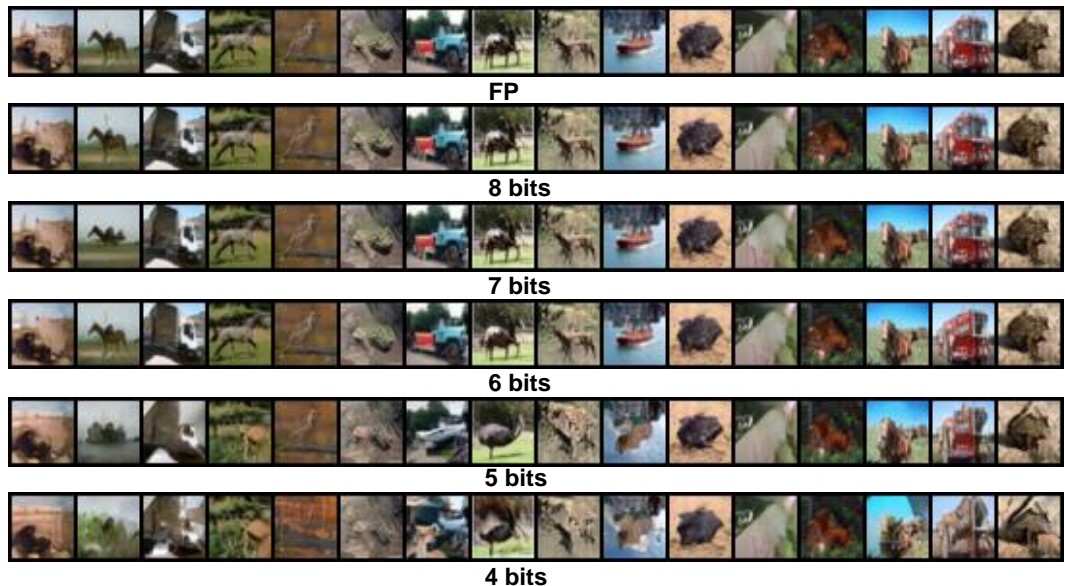

Figure 4: **Random samples from 8/7/6/5/4 bits quantized and full-precision DDPM on CIFAR-10. The resolution of each sample is 32×32.**

On the LSUN-Church dataset, a similar trend is observed as with LSUN-Bedroom. The 6-bit quantization is the lowest precision at which the quality of generated images remains acceptable when applying our quantization method to the DDIM model. At this bit-width, the generated images still retain sufficient detail and visual coherence. However, reducing the bit-width further leads to a noticeable decline in quality, with prominent artifacts and distortions. This indicates that 6 bits represent a critical threshold for maintaining image fidelity in complex datasets like LSUN-Church, highlighting the importance of balancing quantization efficiency with visual performance.

For the experiment on ImageNet, we utilize the Latent Diffusion Model (LDM) as the generation framework. Image generation results are showcased for the LDM model quantized to 8, 6, and 5 bits using our proposed quantization method. Observably, the quality of generated images remains comparable to those produced by the FP model down to 6 bits. This demonstrates the effectiveness of our quantization approach in preserving image fidelity while significantly reducing computational requirements. However, at 5 bits, some image details are lost, indicating a lower bound for maintaining comparable performance.

### A.3    EFFECTIVENESS OF NEW DISTORTION METRICS

As mentioned earlier, we adopted a new metric (MSE+SSIM) instead of solely relying on MSE to measure distortion when allocating different bit-widths across layers. To demonstrate the effectiveness of this newly designed distortion metric, we conducted an ablation study comparing the performance of our method using MSE alone versus the new metric on the CIFAR-10 dataset. The results, shown in Table 3, reveal that applying the combined MSE+SSIM metric led to lower FID scores for generated images when the DDPM model was quantized to 7, 6, 5, and 4 bits, compared to using MSE as the sole metric. This improvement highlights the advantage of incorporating SSIM to capture structural similarity alongside MSE, resulting in better visual quality during quantization.

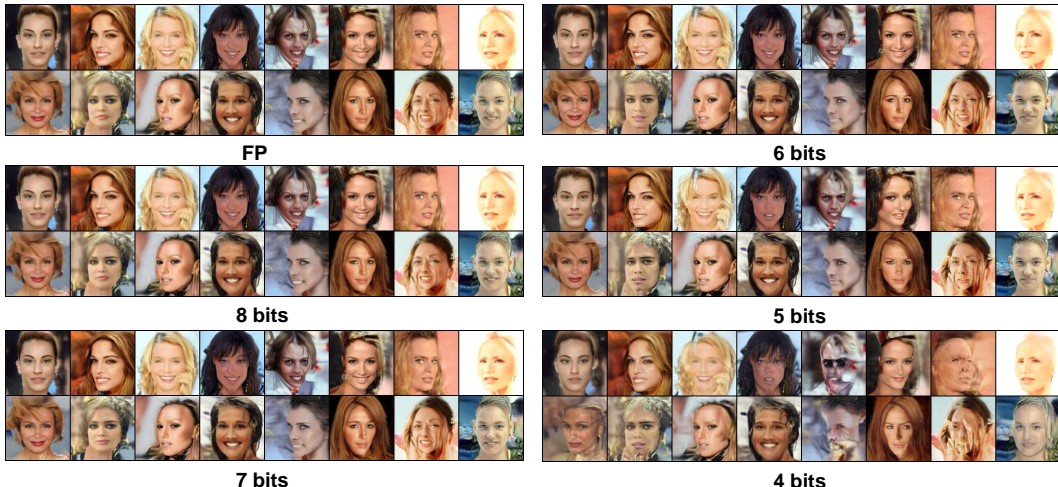

Figure 5: **Random samples from 8/7/6/5/4 bits quantized and full-precision DDPM on Celeba-HQ. The resolution of each sample is 256×256.**

Table 3: Ablation study on the effectiveness of the new metric in improving the performance of 4/5/6/7/8-bit quantized DDPM models on the CIFAR-10 dataset.

|  | Bit-width | IS↑ | IS(std)↓ | FID↓ |
|---|---|---|---|---|
|  | FP | 9.47 | 0.121 | 3.30 |
| **MSE** | 8bits | 9.51 | 0.067 | 3.13 |
|  | 7bits | 9.51 | 0.060 | 3.23 |
|  | 6bits | 9.46 | 0.088 | 3.42 |
|  | 5bits | 8.99 | 0.086 | 6.39 |
|  | 4bits | 6.69 | 0.106 | 34.88 |
| **MSE + SSIM** | 8bits | 9.48 | 0.094 | 3.16 |
|  | 7bits | 9.43 | 0.117 | **3.17** |
|  | 6bits | 9.29 | 0.147 | **3.37** |
|  | 5bits | 9.01 | 0.132 | **6.13** |
|  | 4bits | 6.76 | 0.056 | **33.03** |

## A.4  IMPACT OF QUANTIZATION ACROSS DIFFERENT MODEL LAYERS

Figure 9 demonstrates the bit allocation across different layers of the DDPM model on the CIFAR-10 dataset, showcasing the impact of our proposed quantization method. We observe that with an average bit allocation of 8 bits, some layers are quantized to 10 bits, while others are quantized to 8, 6, or even 4 bits. When the average bit allocation is 4 bits, the quantized bit values across layers range from 10 to 2 bits, including 8, 6, and 4 bits. For certain layers, such as the 36th and 39th, no bits are allocated.

## A.5  RESULTS OF QUANTIZING ON WEIGHTS

We present FID score for weight quantization while maintaining full precision for activations in DDPM models on the CIFAR-10 and CelebA-HQ datasets in Tab. 4. we observe that quantizing only the weights allows parameters to be reduced to 6 bits without significant accuracy degradation (loss < 0.2) on the CIFAR-10 dataset. Similarly, for DDPM on the CelebA-HQ dataset, weight quantization to 6 bits achieves comparable performance, with accuracy loss remaining below 0.6.

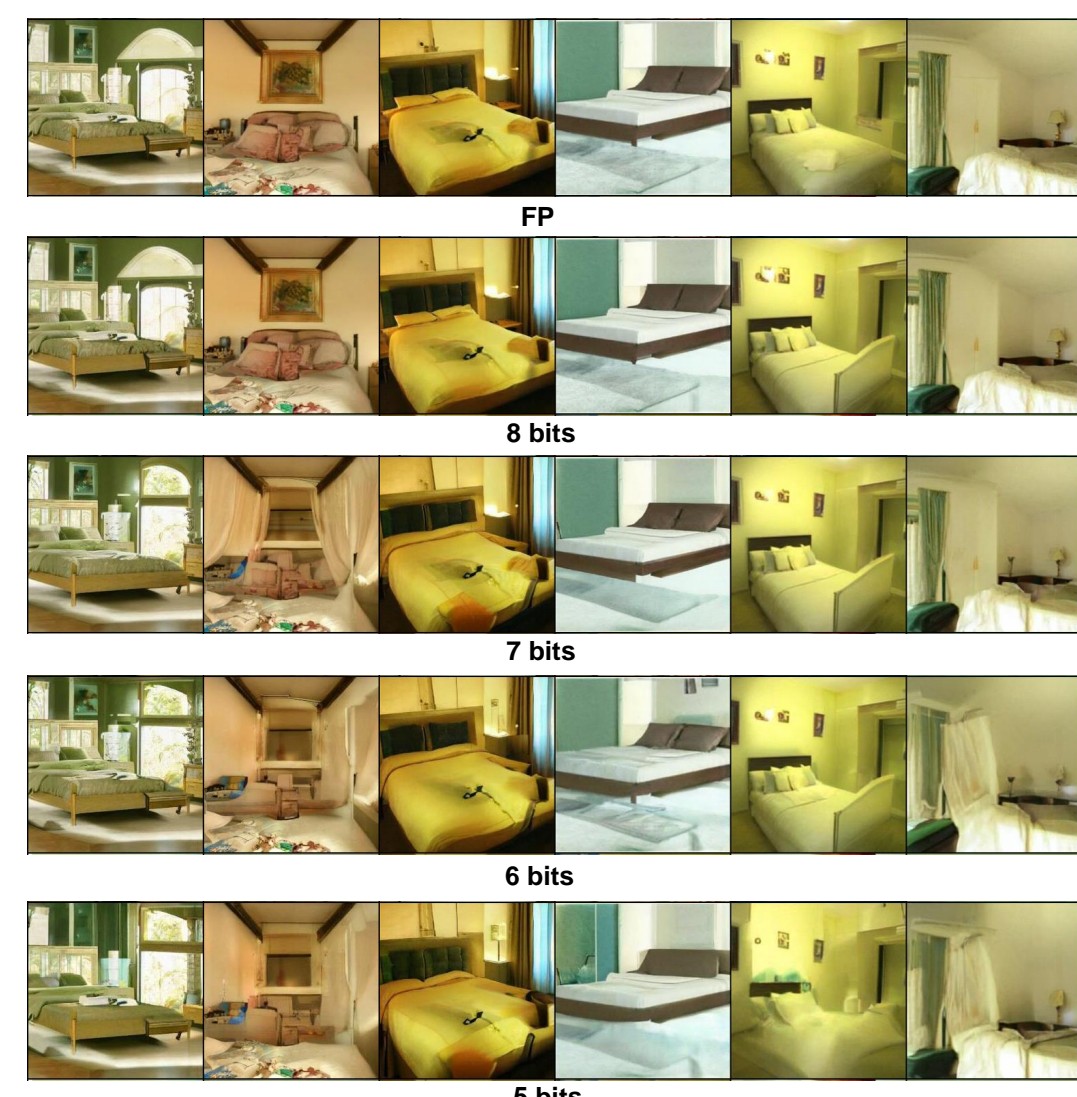

Figure 6: **Random samples from 8/7/6/5 bits quantized and full-precision DDIM on LSUN-Bedroom. The resolution of each sample is 256×256.**

Table 4: Results (FID) of quantization on weights of DDPM model across CIFAR-10 and CeleBA-HQ datasets.

|  | CIFAR-10 | CeleBA-HQ |
| --- | --- | --- |
| FP | 3.30 | 9.01 |
| w=8, a=32 | 3.14 | 8.87 |
| w=7, a=32 | 3.15 | 8.98 |
| w=6, a=32 | 3.30 | 9.04 |
| w=5, a=32 | 6.09 | 9.27 |
| w=4, a=32 | 32.27 | 9.45 |

## A.6 DISCUSSION OF THE NUMBER OF TIMESTEP GROUPS

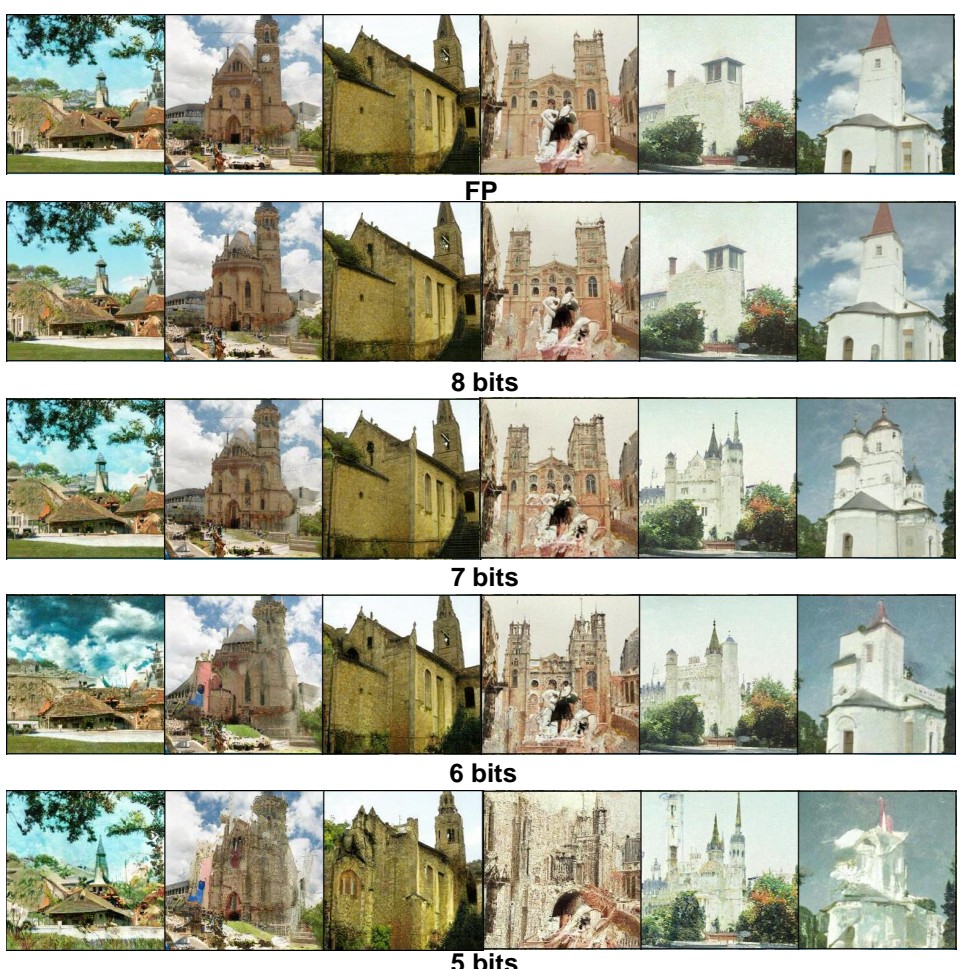

Figure 7: **Random samples from 8/7/6/5 bits quantized and full-precision DDIM on LSUN-Church. The resolution of each sample is 256×256.**

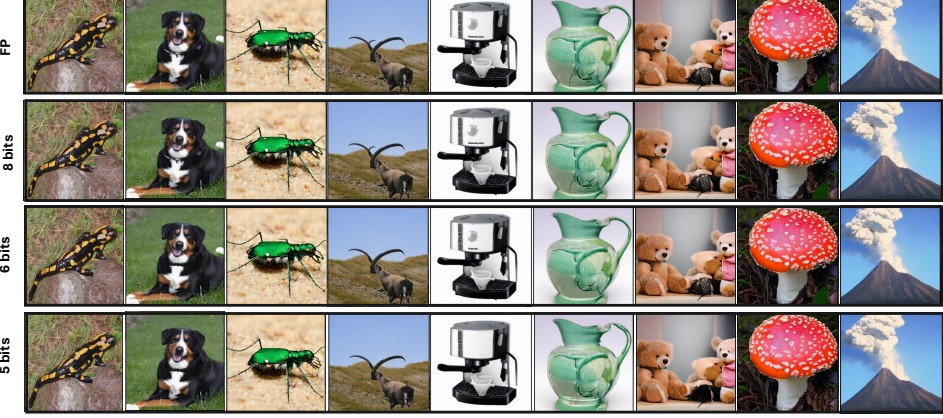

Figure 8: **Random samples from 8/6/5 bits quantized and full-precision LDM on ImageNet. The resolution of each sample is 256×256.**

Table 5 presents the FID scores of quantized DDPM model on the CIFAR-10 dataset with bit allocations determined using distortion computations from two groups of time steps (e.g., the first 100

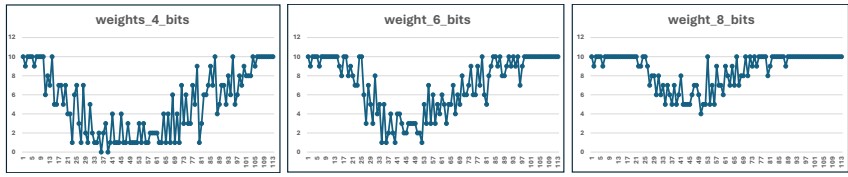

Figure 9: **Visualization of bit allocation across different layers of the DDPM model applied to the CIFAR-10 dataset.**

Table 5: Results (FID) of quantization results with different group numbers.

| Bit Width | 2 Groups (500 timesteps per group) | 10 Groups (100 timesteps per group) |
|---|---|---|
| 8 bits | 58.58 | 45.44 |
| 7 bits | 65.94 | 55.72 |
| 6 bits | 74.46 | 66.39 |
| 5 bits | 95.05 | 83.22 |
| 4 bits | 105.88 | 92.62 |

and 500 time steps) and ten groups of time steps (e.g., the first 100, 200, ..., 1000 time steps). We can see that using more groups of time steps, we can obtain a lower FID score.

## A.7   THE IMPLEMENTATION DETAILS OF OPTIMIZATION

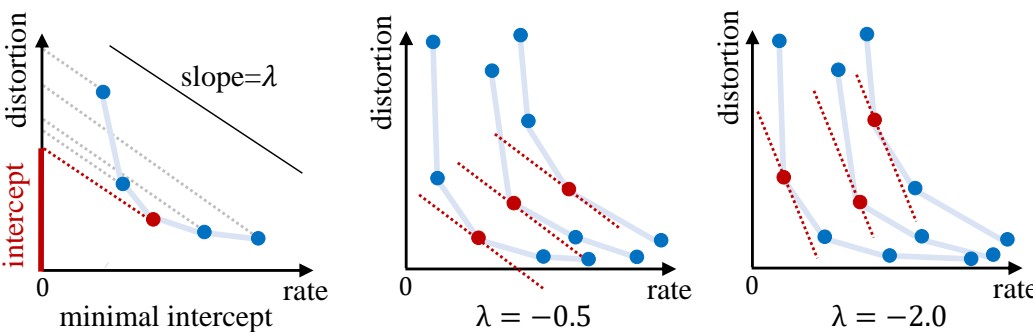

Figure 10: Examples of the optimization method. In the left figure, the red point has minimal intercept on Y-Axis and it is selected. The middle and right figures show the selected points on three curves when $\lambda = -0.5$ and $\lambda = 2.0$, respectively.

We apply Lagrangian formulation to solve objective function (13). According to the additivity property, the Lagrangian cost function of (13) is defined as

$$\sum_{i=1}^{l} \Gamma_{\mathbf{W}_i^\dagger}(\mathbf{O}, \widehat{\mathbf{O}}) + \sum_{i=1}^{l} \Gamma_{\mathbf{I}_i^\dagger}(\mathbf{O}, \widehat{\mathbf{O}}) - \gamma \cdot \left(\sum_{i=1}^{l} s_{\mathbf{W}_i^\dagger} + \sum_{i=1}^{l} s_{\mathbf{I}_i^\dagger}\right), \tag{16}$$

in which $\gamma$ decides the trade-off between bit rate and output distortion. Setting the partial derivations of (16) to zero with respect to each $s_{\mathbf{W}_i^\dagger}$ and $s_{\mathbf{I}_i^\dagger}$, we obtain the optimal condition

$$\frac{\partial \Gamma_{\mathbf{W}_i^\dagger}(\mathbf{O}, \widehat{\mathbf{O}})}{\partial s_{\mathbf{W}_i^\dagger}} = \frac{\partial \Gamma_{\mathbf{I}_i^\dagger}(\mathbf{O}, \widehat{\mathbf{O}})}{\partial s_{\mathbf{I}_i^\dagger}} = \gamma. \tag{17}$$

for all $1 \leq i \leq l$. Equation (17) expresses that the slopes of all rate-distortion curves (output distortion versus bit rate) should be equal to obtain optimal bit allocation with minimal output distortion. According to (17), we are able to solve objective function (13) efficiently by enumerating slope $\gamma$ and then choosing the point on each rate-distortion curve with slope equal to $\gamma$ as the solution.

---

**Algorithm 1** Generating rate-distortion curves for the weights and activations of each layer

---

**Input:** Diffusion Model $\mathcal{F}$; Input $\mathbf{I}$.
**Output:** Rate-distortion curves: $Q_1, Q_2, ..., Q_{2l}$.
 1: Compute original output vector $\mathbf{Y} = \mathcal{F}(\mathbf{I})$.
 2: **for** each group $\mathbf{C}_i$ (weights or activations in a layer) **do**
 3:    **for** bit-width $b$ ranging from 1 bit to C bits **do**
 4:       Quantize $\mathbf{C}_i$ with $b$ bits: $\mathbf{C}_i = q(\mathbf{C}_i)$.
 5:       Compute the size $R$ of quantized $\mathbf{C}_i$, $q(\mathbf{C}_i)$.
 6:       Compute modified output vector $\widehat{\mathbf{Y}}$.
 7:       Compute output distortion $d = distance(\mathbf{Y}, \widehat{\mathbf{Y}})$, which is SSIM + MSE.
 8:       Collect point $P = (R, d)$.
 9:       Update $Q_i = Q_i \cup P$.
10:    **end for**
11: **end for**

---

**Algorithm 2** Optimization via Lagrangian formulation

---

**Input:** Rate-distortion curves: $Q_1, Q_2, ..., Q_{2l}$; Slope $\gamma$.
**Output:** Solution of optimal bit allocation $S$; Size of quantized model $R$.
 1: **Initialize** $S = \emptyset$, $R = 0$.
 2: **for** each rate-distortion curve $Q_i$ **do**
 3:    **Initialize** $Y_{min\_intercept} = \infty$, $id = -1$.
 4:    **for** each point $P$ on $Q_i$ **do**
 5:       $x_0 = P_j \rightarrow x$, $y_0 = P_j \rightarrow y$, $Y_{intercept} = y_0 - \lambda \cdot x_0$.
 6:       **if** $Y_{intercept} < Y_{min\_intercept}$ **then**
 7:          **Update** $Y_{min\_intercept} = Y_{intercept}$.
 8:          **Update** $id = j$.
 9:       **end if**
10:    **end for**
11:    **Update** $S = S \cup \{P_{id}\}$, $R = R + P_{id} \rightarrow x$.
12: **end for**

---

The algorithm works as follows. Before optimization, we quantize weights and activations of each layer with different bit widths and calculate the output distortion caused by quantization to generate the rate-distortion curve for each layer's weights and activations. After that, we assign a real value to $\gamma$, and select the point with slope equal to $\gamma$ on each curve. The selected points on all curves correspond to a group of solution for bit allocation. In practice, we explore multiple values for $\gamma$ until the size of the quantized network meets constraint. We randomly select 50 images from ImageNet dataset to calculate output distortion caused by quantization. Assume that we have $N$ curves and $M$ points in each curve. The time complexity to find optimum bit allocation is $O(K \cdot M \cdot N)$, where $K$ is the total number of slope $\gamma$ to be evaluated, $M$ is the total number of bit widths, and $N$ is the total number of layers.

Figure 10 illustrates an example of the optimization method. The optimization starts from enumerating the value $\gamma$. Given a $\gamma$, we find the point with the slope equal to $\gamma$ on each curve. Specifically, we compute the intercepts on Y-axis for all the lines with slope $\lambda$ passing one of the points on the curve. The point on the curve passed by the line with minimal intercept on Y-axis is selected. For the selected point, the horizontal coordinate (the value in the X-axis direction) corresponds to the bit width of the weights or activations in the layer. Algorithm 1 and 2 show the pseudo-codes to generate rate-distortion curves and to optimize the bit allocation.

