# OpenReview forum: "Distribution-Aware Diffusion Model Quantization via Distortion Minimization"
_ICLR.cc/2025/Conference — Submitted to ICLR 2025_

### Official Review · Reviewer_KDiN · 2024-10-31

**Soundness:** 2
**Presentation:** 2
**Contribution:** 2
**Rating:** 5
**Confidence:** 4

**Summary:**

This paper proposes a post-training quantization (PTQ) approach to reduce the model size and computational complexity of diffusion models. Specifically, it introduces a method that splits timesteps into groups, optimizing quantization parameters within each group independently. The quantization process for each group is then formulated as a rate-distortion optimization problem to minimize output distortion. Additionally, the approach employs mixed-precision quantization, applying different bit widths across layers.

**Strengths:**

1. Quantization in the diffusion model is important.
2. The use of mixed-precision quantization, with varying bit widths across layers, enhances the adaptability and precision of the quantization process.
3. Formulating the quantization as an optimization problem is reasonable.

**Weaknesses:**

1. Please elaborate on the advantages and differences of your mixed-precision strategy compared to prior quantization methods like MixDQ [1] and BitsFusion [2]. And provide a brief comparison table or paragraph highlighting key differences in approach and results.

2. The "Related Works" section has numerous typos and informal language, such as in lines 122 and 128 where "Zhong et al. Zhong et al. (2022)" and "Ma et al. Ma et al. (2023)" are repeated. And I also recommend consolidating several paragraphs into one paragraph in Section 2.2 for better flow.

3. Additionally, some sentences in this section are confusing. For instance, "Li et al. introduced a post-training quantization (PTQ) Li et al. (2023b) method called Q-Diffusion, tailored specifically to the distinctive multi-timestep pipeline and model architecture of diffusion models," could be rephrased for clarity. For example, " Li et al. (2023b) proposes Q-Diffusion, which is a PTQ method tailored specifically to the distinctive multi-timestep pipeline and model architecture of diffusion models". The preliminaries would also benefit from conciseness due to limited space, especially for details on diffusion models. For example, Eq. (2) and (4) can be removed or inserted in the paragraph.

4. The motivation for splitting timesteps into groups needs clarification. Currently, it's unclear how the approach of “splitting timesteps into groups to optimize quantization for each group” logically leads to “using mixed precision to quantize parameters across layers.” in line 215. Authors could include a brief explanation or diagram illustrating how these components of their approach are connected.

5. Lastly, Table 2 does not sufficiently demonstrate the advantages of the proposed method. With recent works achieving quantization at or below 4 bits, such as MixDQ (4 bits) and BitsFusion (1.99 bits), the lowest comparison at 6 bits in Table 2 is less compelling. Authors can include results for lower bit-widths (e.g. 2 bits, 4 bits) in their comparisons or explain why their method may not be applicable at such low bit-widths if that is the case. Additionally, it's better to provide more results compared to state-of-the-art methods [1][2].

[1] MixDQ: Memory-Efficient Few-Step Text-to-Image Diffusion Models with Metric-Decoupled Mixed Precision Quantization. ECCV 2024.

[2] BitsFusion: 1.99 bits Weight Quantization of Diffusion Model. NeurIPS 2024.

**Questions:**

1. What is the difference of your mixed-precision strategy compared to prior quantization methods like MixDQ [1] and BitsFusion [2].

[1] MixDQ: Memory-Efficient Few-Step Text-to-Image Diffusion Models with Metric-Decoupled Mixed Precision Quantization. ECCV 2024.

[2] BitsFusion: 1.99 bits Weight Quantization of Diffusion Model. NeurIPS 2024.

2. What is the motivation for splitting timesteps into groups?

---

> ### Author Response · Authors · 2024-11-25
> **Response to Reviewer KDiN's Comments**
>
> We deeply thank Reviewer KDiN for the careful review and the constructive suggestions. Below please check our answers to the questions.
>
> Q1. Please elaborate on the advantages and differences of your mixed-precision strategy compared to prior quantization methods like MixDQ [1] and BitsFusion [2]. And provide a brief comparison table or paragraph highlighting key differences in approach and results.
>
> Ans: Our approach is different with MixDQ and BitsFusion in several aspects, which are listed in Table 1 below. First, both MixDQ and BitsFusion focus on the quantization of text-to-image diffusion models and the results they reported are all on text-to-image datasets. Our approach focuses on the quantization of image-to-image diffusion models and we reported results on the related datasets. Besides, our approach applied a different method to design the quantization scheme. MixDQ and BitsFusion applied sensitivity metrics to decide the quantization bit widths of the layers, while our approach formulated the optimal bit widths of the layers as the minimization of the output error. What’s more, our approach adopted a different optimization method. MixDQ and BitsFusion adopted integer programming and greedy algorithm respectively for optimization, while our approach adopted Lagrangian formulation for optimization. The solution obtained by MixDQ and BitsFusion is locally optimal. Our approach can obtain globally optimal solution using the Lagrangian formulation. One thing worth to mention is that in BitsFusion, it requires to train the model after quantization, which is time-consuming and has high time complexity. This is the reason why this approach can quantize the model into very small bit widths (1.99 bits). Different with BitsFusion, both MixDQ and our approach are post-training quantization, which does not train the model after quantization. We have also added the discussion of MixDQ and BitsFusion in the manuscript. Thanks for pointing out the two important works.
>
> Table 1 - Comparison with MixDQ and BitsFusion on Multiple Aspects.
>
> |            | Model                     | Method                    | OPtimization           | Solution | Training |
> |------------|---------------------------|---------------------------|------------------------|----------|----------|
> | MixDQ      | Text-to-image generation  | Sensitivity metrics       | Interger programming   | Local    | No       |
> | BitsFusion | Text-to-image generation  | Sensitivity metrics       | Greedy algorithm       | Local    | Yes      |
> | Ours       | Image-to-Image generation | Output error minimization | Lagrangian formulation | Global   | No       |
>
>
> Q2. The "Related Works" section has numerous typos and informal language, such as in lines 122 and 128 where "Zhong et al. Zhong et al. (2022)" and "Ma et al. Ma et al. (2023)" are repeated. And I also recommend consolidating several paragraphs into one paragraph in Section 2.2 for better flow.
>
> Ans: We corrected the typos in line 122, line 128, and other lines in the related work section. We also consolidated the paragraphs in Section 2.2. Thanks a lot for the careful review and the suggestion.
>
>
> Q3. Additionally, some sentences in this section are confusing. For instance, "Li et al. introduced a post-training quantization (PTQ) Li et al. (2023b) method called Q-Diffusion, tailored specifically to the distinctive multi-timestep pipeline and model architecture of diffusion models," could be rephrased for clarity. For example, " Li et al. (2023b) proposes Q-Diffusion, which is a PTQ method tailored specifically to the distinctive multi-timestep pipeline and model architecture of diffusion models". The preliminaries would also benefit from conciseness due to limited space, especially for details on diffusion models. For example, Eq. (2) and (4) can be removed or inserted in the paragraph.
>
> Ans: We revised the sentence “Li et al. introduced a post-training quantization (PTQ) Li et al. (2023b) method called Q-Diffusion, tailored specifically to the distinctive multi-timestep pipeline and model architecture of diffusion models” to the sentence corrected by the reviewer. We inserted Equation (2) in the text and removed Equation (4) due to the limited space. Thanks a lot for the suggestion.

---

> ### Author Response · Authors · 2024-11-25
> **Response to Reviewer KDiN's Comments - Part 2**
>
> Q4. The motivation for splitting timesteps into groups needs clarification. Currently, it's unclear how the approach of “splitting timesteps into groups to optimize quantization for each group” logically leads to “using mixed precision to quantize parameters across layers.” in line 215. Authors could include a brief explanation or diagram illustrating how these components of their approach are connected.
>
> Ans: Because the distributions of activations are different across timesteps, we thus divide the timesteps into groups and optimize the quantization of each group separately. Then for each of the groups, we apply mixed precision and find the optimal bit widths for the layers, given the fact that different layers react differently to quantization. We added the explanation of the splitting timesteps and mixed precision in the manuscript. Thanks for this suggestion.
>
>
> Q5. Lastly, Table 2 does not sufficiently demonstrate the advantages of the proposed method. With recent works achieving quantization at or below 4 bits, such as MixDQ (4 bits) and BitsFusion (1.99 bits), the lowest comparison at 6 bits in Table 2 is less compelling. Authors can include results for lower bit-widths (e.g. 2 bits, 4 bits) in their comparisons or explain why their method may not be applicable at such low bit-widths if that is the case. Additionally, it's better to provide more results compared to state-of-the-art methods [1][2].
>
> Ans: MixDQ and BitsFusion focus on the quantization of text-to-image diffusion models. Different with them, our approach focuses on the quantization of image-to-image diffusion models. Note that BitsFusion requires to train the model from scratch after quantization, which is time-consuming, and the method has high time complexity. This is also the reason why BitsFusion can go down to very low bit widths (1.99 bits). Different with BitsFusion, our approach is a post-training quantization approach, which does not require to train the model after quantization. The baseline methods we compared with are all post-training quantization methods. We reported our results at 4 bits on four datasets, which is listed in Table 2 below. As we can see, the performance has dropped noticeably at 4 bits. With the model trained after quantization, the performance can be largely improved. We didn't report the results with training in the manuscript, as this is out of the range of this paper, and we are not able to train the models given the limited time of the rebuttal period. We will add the training results after the training is done in the manuscript. Thanks for this insight.
>
> Table 2 - Results (FID) at 4 bits on different datasets
> | Dataset      | Model        | Full Precision | 4 Bits |
> |--------------|--------------|----------------|--------|
> | CIFAR-10     | DDPM         | 3.30           | 33.03  |
> | Celeba-HQ    | DDPM         | 9.01           | 9.49   |
> | LSUN-Bedroom | LSUN-Bedroom | 7.89           | 39.88  |
> | LSUN-Church  | LSUN-Church  | 11.33          | 59.74  |

---

> ### Author Response · Authors · 2024-11-27
>
> Dear Reviewer KDiN,
>
> We appreciate it if you could let us know whether our responses are able to address your concerns. We are happy to address any further concerns. Thank you.
>
> Sincerely.

---

> ### Author Response · Authors · 2024-12-03
>
> Dear Reviewer KDiN,
>
>
> We answered the questions in your comments and revised the manuscript accordingly, including:
> 1. Elaborated on the advantages of our proposed mixed-precision quantization approach, discussed the differences with prior works (MixDQ and BitsFusion), and added a comparison table to highlight the key differences.
> 2. Fixed the typos in the related work section, and consolidated the paragraphs in Section 2.2.
> 3. Rephrased the confusing sentences in the related work section, and removed the equations in the preliminaries section.
> 4. Explained the motivation for splitting the time steps and the connections between the components of our approach.
> 5. Added the results at the low bit width (4 Bits).
>
> Note that for MixDQ and BitsFusion, they focus on the quantization of text-to-image diffusion models. In this paper, our approach and the baseline methods we compared with focus on the quantization of image-to-image diffusion models. BitsFusion required to train the model from scratch, which is the reason that this method can quantize the model into very low bit width (1.99 bits). Different with BitsFusion, our approach is a post-training quantization method and does train or re-train the model after quantization. We discussed this in our answers. Please check our answers for more details.
>
> We appreciate if you can let us know whether our answers have addressed your concerns or if you have other questions. We are glad to address your further questions and concerns.
>
> Sincerely

---

### Official Review · Reviewer_9DVu · 2024-11-01

**Soundness:** 3
**Presentation:** 3
**Contribution:** 3
**Rating:** 8
**Confidence:** 2

**Summary:**

This paper proposes a post-training quantization approach for diffusion models. The paper finds that distributions of the outputs of diffusion models differ significantly across timesteps, and they utilize this finding to split the timesteps into meaningful groups and optimize the quantization configuration of each group separately. Empirical results demonstrate that the proposed approach achieves significant improvements over state-of-the-art methods, enabling the reduction of diffusion models' bit width to 5-6 bits while preserving high accuracy.

**Strengths:**

- The paper is written clearly and is easy to read.
- The paper provides both theoretical as well as empirical evidence that the proposed approach works.

**Weaknesses:**

For details, please see the Questions section.
- The motivation behind some of the design choices are not explained clearly
- Some parts of the proposed method is missing elaboration

Minor:
- The paper does not provide specific implementation details such as the exact algorithms or code snippets used for quantization and optimization. Including these would help in reproducing the results.
- Including a user study or qualitative analysis to assess the perceptual quality of the generated images could provide additional insights into the effectiveness of the quantization method.

**Questions:**

- It is not clear how the timesteps are grouped. Please provide more details, as this is one of the major steps in the algorithm.
- It is mentioned that “the output distortion is defined as the Structural Similarity Index Measure (SSIM) loss between $O$ and $\hat{O}$ plus the MSE”. Is this a common method for evaluating output distortion? Why not e.g., PSNR?

---

> ### Author Response · Authors · 2024-11-23
> **Response to Reviewer 9DVu's Comments**
>
> We deeply thank Reviewer 9DVu for the careful review and the constructive suggestions. Below please check our answers to the questions.
>
> Q1. The paper does not provide specific implementation details such as the exact algorithms or code snippets used for quantization and optimization. Including these would help in reproducing the results.
>
> Ans: We added the implementation details of the optimization algorithm including the pseudo code in the appendix. Specifically, according to the optimal condition in Equation (18), we can find the optimal bit allocation when the slopes of all output error versus size curves are equal. As a result, we can solve the optimization problem by enumerating the slope value $gamma$ and choosing the point with slope equal to $gamma$ on each of the curves. We will also release our source code in the finalized manuscript. Thanks for pointing out this important question.
>
> Q2. Including a user study or qualitative analysis to assess the perceptual quality of the generated images could provide additional insights into the effectiveness of the quantization method.
>
> Ans: Thanks for this question. We illustrated the qualitative results on four datasets in the appendix, including CIFAR-10, Celeba-HQ, LSUN-Bedroom, and LSUN-Church. Besides, in the rebuttal period, we also added the image generation results on the large scale ImageNet dataset, which are shown in Figure 8 in the Appendix. For the experiment on ImageNet, we applied the Latent Diffusion Model (LDM) as the generation framework, and the image generation results are showcased for the LDM model quantized to 8, 6, and 5 bits using our proposed quantization method. On ImageNet, the quality of generated images remains comparable to those produced by the full-precision model when the model is quantized down to 6 bits. This also demonstrates the effectiveness of our quantization approach in preserving image fidelity while significantly reducing computational requirements.
>
> Q3. It is not clear how the timesteps are grouped. Please provide more details, as this is one of the major steps in the algorithm.
>
> Ans: We divided the time steps into 10 groups with equal size. Specifically, there are totally 1000 time steps where they are equally divided into 10 groups and each group has 100 time steps. Dividing the time steps into more groups can further improve the accuracy, as in this case we can do more fine-grained quantization, but it may also significantly increase the time complexity of the method. Table 1 below illustrates the results with different numbers of groups (2 groups and 10 groups). As we can see, the accuracy is improved noticeably when increasing the group number from 2 to 10. In practice, we divide the time steps into 10 groups after considering both the accuracy and the algorithm’s time complexity. We added the discussion of the time steps in Section A6  in the appendix. Thanks for this constructive suggestion.
>
> Table 1 - Results (FID) of quantization results with different group numbers
>
> | Bit Width | 2 Groups (500 timesteps per group) | 10 Groups (100 timesteps per group) |
> |-----------|------------------------------------|-------------------------------------|
> | 8 bits    | 58.58                              | 45.44                               |
> | 7 bits    | 65.94                              | 55.72                               |
> | 6 bits    | 74.46                              | 66.39                               |
> | 5 bits    | 95.05                              | 83.22                               |
> | 4 bits    | 105.88                             | 92.62                               |
>
>
> Q4. It is mentioned that “the output distortion is defined as the Structural Similarity Index Measure (SSIM) loss between O  and O^  plus the MSE”. Is this a common method for evaluating output distortion? Why not e.g., PSNR?
>
> Ans: MSE is widely used in prior works to measure the quantization results. The PSNR is highly related to MSE. It is actually the log representation of MSE, i.e. PSNR = 10 log (L^2/MSE) where L is the largest value. Our approach applies the Structural Similarity Index Measure (SSIM) added by the Mean Squared Error (MSE) to define output distortion, where SSIM indicates the picture level similarity and MSE indicates pixel level similarity. As a result, our method takes the similarity in both picture level and pixel level into consideration.

---

> ### Author Response · Authors · 2024-11-27
>
> Dear Reviewer 9DVu,
>
> We appreciate it if you could let us know whether our responses are able to address your concerns. We are happy to address any further concerns. Thank you.
>
> Sincerely.

---

### Official Review · Reviewer_STQM · 2024-11-03

**Soundness:** 1
**Presentation:** 2
**Contribution:** 2
**Rating:** 3
**Confidence:** 4

**Summary:**

The authors present an approach to jointly quantize weights and activations of a diffusion model after it has been trained (i.e, post training quantization). To do so, a joint optimization problem is formuled: the authors propose to minimize the structured similarity index between original and quantized model, where the optimization paramters are bitwidth of quantization of weights and activations. The authors show that global SSIM optimization can be simplified (under some assumptions) into sum over SSIM measurements after every layer.

**Strengths:**

-

**Weaknesses:**

In the current form paper does not present the complete information to assess its merits and contributions. Paper's writing (grammar, style, flow) is good, but once it comes to describing the actual working soluiton (end of section 3) it strugles to present the final algorithm. There are inconsistentices in what is announced (i.e., abstract/intro) vs what is presented and demonstrated (experiments). The issue is significant: I am not confident that any reader would be able to implement the approach following the presentation in the paper.  I am not able to give any positive rating to this paper while these issues are clarified and/or resolved.

The authors present an approach to jointly quantize weights and activations of a diffusion model after it has been trained (i.e, post training quantization). To do so, a joint optimization problem is formuled: the authors propose to minimize the structured similarity index between original and quantized model, where the optimization paramters are bitwidth of quantization of weights and activations. The authors show that global SSIM optimization can be simplified (under some assumptions) into sum over SSIM measurements after every layer.

Unfortunately, this is as much details I can provide about their approach after reading the paper. Paper omits the final presentation of the quantization approach/algorithm; has some logical inconsistencies, and as such raises many questions:
1. Section 3.4 does not present any readily awailable optimization algorithm. Lines 350-357 describe an approach, but it is impossible to follow.
2. Given the problem defintion (eq.8, eq.16), authors never discuss what kind of optimization it is: in my understanding it is mixed-integer (some parts are fully differentiably, some parts are integer, like sizes of weights/activations). Therefore, in the same section 3.4 I do not understand how authors propose taking a derivative wrt s_w; which is defined as size of the weights?
3. When describing the soltuion on lines 350-357; authors propose enumeration over different choices; it would be highly benefitial to make very presize statements: what is being enumertaed, what are different choices there etc. As such, saying that total runtime of O(K*M*N) has "linear time complexity" is misleading: linear wrt what? size of the model? if it is then K, M, N needs to be clearly counted
4. The authors write that algorithm uses mixed precision (layers can have different bitwidth) and timestep-aware quantization. There is no word on how these are achieved and no experimental evaluation/discussion.
5. Experiments. I would like to understand how big are the networks that are being compressed, what is their architecture, and other basic information. Has authors trained them from scratch or obtained from somewhere? Visually speaking, it seems like 6bits quantizaiton introduces noticable quality degradation, and probably shouldn't be counted as a major achievement.
6. Experimetns. Where are the experimetnts where per-layer bitwidt are different? All presented experiments have same per layer setting.

**Questions:**

Please see weaknesses above.

---

> ### Author Response · Authors · 2024-11-27
> **Response to Reviewer STQM's Comments**
>
> We deeply thank Reviewer STQM for the careful review and the constructive suggestions. Below please check our answers to the questions.
>
> Q1. Section 3.4 does not present any readily awailable optimization algorithm. Lines 350-357 describe an approach, but it is impossible to follow.
>
>
> Ans: We added a section (Section A.7 in the Appendix) to explain the implementation details of the optimisation algorithm and show the seudo-codes of the algorithm. We will also release the link of the source codes of our approach in the finalised manuscript.
>
> Specifically, we apply Lagrangian formulation to solve objective function in Equation 13. According to the additivity property, we can obtain the optimal condition in Equation 15, which expresses that the slopes of all rate-distortion curves (output distortion versus size) should be equal to obtain optimal bit allocation with minimal output distortion. We can then solve the objective function in Equation 13 efficiently by enumerating slope  $\gamma$ and then choosing the point on each rate-distortion curve with slope equal to $\gamma$ as the solution.
>
> The algorithm works as follows.
> Before optimization, we quantize weights and activations of each layer with different bit widths and calculate the output distortion caused by quantization to generate the rate-distortion curve for each layer's weights and activations.
> After that, we assign a real value to $\gamma$, and select the point with slope equal to $\gamma$ on each curve.
> The selected points on all curves correspond to a group of solution.
> We added the implementation details and the seudo-codes of the algorithm in Section A.7. Thanks for this constructive suggestion.
>
>
> Q2. Given the problem defintion (eq.8, eq.16), authors never discuss what kind of optimization it is: in my understanding it is mixed-integer (some parts are fully differentiably, some parts are integer, like sizes of weights/activations). Therefore, in the same section 3.4 I do not understand how authors propose taking a derivative wrt s_w; which is defined as size of the weights?
>
> Ans: Our approach applied the method of Lagrange multipliers (Lagrangian formulation) to do the optimization. According the definition of Lagrangian formulation: in order to find the maximum or minimum of a function $f(x)$ subject to constraint $g(x)$, it should find the stationary points of $L = f(x) + \lambda \cdot g(x)$. This means that all partial derivatives of $L$ should be zero. In our case, $f(x)$ is the learning objective - minimizing output error of the quantised model, $g(x)$ is the contraint of the size of quantized weights and activations, and the variables are the sizes of weights and activations ($s_W$, $s_I$). As a result, we take the all partial derivatives of the variables ($s_W$, $s_I$) and set the partial derivatives to zero to find the solution.
>
>
> Q.3 When describing the soltuion on lines 350-357; authors propose enumeration over different choices; it would be highly benefitial to make very presize statements: what is being enumertaed, what are different choices there etc. As such, saying that total runtime of O(KMN) has "linear time complexity" is misleading: linear wrt what? size of the model? if it is then K, M, N needs to be clearly counted
>
> Ans: We enumerate slope $\gamma$ in Equation 15 to solve the optimization problem. Equation 15 expresses the optimal condition that  the slopes of all rate-distortion curves (output distortion versus size) should be equal. As a result, we can find the minimal solution by enumerating slope $\gamma$ and choosing the point on each rate-distortion curve with slope equal to $\gamma$. The time complexity of our optimization algorithm is $O(K \cdot M \cdot N)$, where $K$ is the total number of slope $\gamma$ to be evaluated, $M$ is the total number of bit widths, and $N$ is the total number of layers.

---

> ### Author Response · Authors · 2024-11-27
> **Response to Reviewer STQM's Comments - Part 2**
>
> Q4. The authors write that algorithm uses mixed precision (layers can have different bitwidth) and timestep-aware quantization. There is no word on how these are achieved and no experimental evaluation/discussion.
>
> Ans: We added the analysis of the bit allocation across layers when the model is quantized to a given size in Section A.4 in the appendix. Usually, different layers react differently to quantization. Some layers are sensitive to quantization. Quantizing the sensitive layers to low bit widths can lead to a significant accuracy drop. However, for the layers which are not sensitive to quantization, we can quantize them to relatively low bit widths. We observed that the first layers and the last layers usually receive higher bit widths. This is because first layers are related to the input and last layers are related to the output. Layers in the middle layers receive lower bit widths. The details of the bit allocation across the layers are illustrated in the newly added Figure 9. Thanks for this insight.
>
> We also added the discussion of the impact of the number of time step groups. Dividing the time steps into more groups can further improve the accuracy, as in this case we can do more fine-grained quantization, but it may also significantly increase the time complexity of the method. In practice, we divide the time steps into 10 groups after considering both the accuracy and the algorithm’s time complexity. The detailed results of the impact of the number of time steps are illustrated in Section A6 in the appendix.
>
>
>
> Q5. Experiments. I would like to understand how big are the networks that are being compressed, what is their architecture, and other basic information. Has authors trained them from scratch or obtained from somewhere? Visually speaking, it seems like 6bits quantizaiton introduces noticable quality degradation, and probably shouldn't be counted as a major achievement.
>
>
> Ans: On the CIFAR-10 dataset and Celeba-HQ dataset, we evaluated the Denoising Diffusion Implicit Model (DDIM) architecture where the model size is 3.2GB with 800 million parameters. On the LSUN-Bedroom dataset and LSUN-Church dataset, we evaluated the Denoising Diffusion Probabilistic Model (DDPM) architecture where the model size is 1GB with 250 million parameters. We did not train the models from scratch, but downloaded the pre-trained models from the official Github links. We removed the 6 bits quantization from the major achievement. Apologise for this inappropriate claim.
>
> Q6. Experimetns. Where are the experimetnts where per-layer bitwidt are different? All presented experiments have same per layer setting.
>
> Ans: Different layers may receive different bit widths for quantization. We illustrated the results of the bit allocation across layers in the newly added Figure 9 in the manuscript. Thanks for pointing out this important question.

---

> ### Author Response · Authors · 2024-11-27
>
> Dear Reviewer STQM,
>
> We appreciate it if you could let us know whether our responses are able to address your concerns. We are happy to address any further concerns. Thank you.
>
> Sincerely.

---

> ### Author Response · Authors · 2024-12-03
>
> Dear Reviewer STQM,
>
>
> We answered the questions in your comments and revised the manuscript accordingly, including:
> 1. Presented the implementation details of the optimization algorithm.
> 2. Explained the derivative of the variables.
> 3. Explained the enumeration and the time complexity.
> 4. Discussed the bit allocation of mixed-precision quantization and the impact of the number of time steps.
> 5. Provided the information of the network architecture, model size, and training.
>
> Specifically, in the manuscript, we added Section A.4 to discuss the bit allocation across layers in our mixed precision quantization, Section A.6 to discuss the impact of the number of time steps, and Section A.7 to provide the implementation details of the optimization and show the seudo-codes of the algorithm, and revised the related texts in the manuscript accordingly.
>
> We appreciate if you can let us know whether our answers have addressed your concerns or whether you have other questions. We are glad to address your further questions and concerns.
>
> Sincerely

---

### Official Review · Reviewer_pc5T · 2024-11-04

**Soundness:** 2
**Presentation:** 3
**Contribution:** 1
**Rating:** 6
**Confidence:** 4

**Summary:**

This work focuses on post-training quantization of the diffusion models for both the model parameters and layer activations. It adopts a mixed precision approach wherein different layers are quantized with different number of bits for precision. It groups different diffusion time steps and applies the quantization procedure per group. At the heart of this paper is quantization based on output distortion that is defined by the Structural Similarity Index Measure (SSIM) loss between the original layer output and the quantized layer output. The paper formulates this as an optimization across layers and show an additive property of the output distortion. Further, it splits this optimization on per layer output distortion objective, which is easily tractable. Finally, various empirical evaluations are conducted to quantize diffusion models trained on CIFAR-10, CelebaHQ, LSUN-Bedroom, and LSUN-Church datasets. This work shows performance of models quantized from 8-bit to 4-bit and evaluates them both quantitatively as well as qualitatively.

**Strengths:**

- Post-training quantization results on various datasets shows that these diffusion models can be quantized efficiently to bits as low as 6 bits (both in weights/activations) without loosing lot of quantitative performance.
- Proposed scheme performances quite competitively compared to other post-training frameworks.

**Weaknesses:**

- Lack of insights into how quantization affects different model layers. Since these insights would be very helpful to understand the impact of various components in the diffusion models.
- Since the experiments are done on smaller datasets, it would be hard to understand how the conclusions transfer to other networks / datasets / diffusion frameworks.
- Although the quantitative metrics look good in Table 1, qualitative images in Figure 2 shows some artifacts in 6bit compression. It would be good to evaluate these more thoroughly.

**Questions:**

- Why use the output distortion metric (aka SSIM between output of the original and quantized models)? Since there are many other ways to define the distance between these two outputs (like mean-squared error, maximum mean discrepancy, absolute-difference, or other form of distributional distance metric)?
- In Eq.7, why does the total size constraint sum up both the quantized activation and the quantized weights? Shouldn't the weights and activations have separate constraint, one for the model parameters and other for the activations?
- Have you tried only parameter quantization to see how much lower bit-widths the models can tolerate if activations are kept in higher bits (8bit or higher)?
- How does one adapt the proposed quantization scheme for quantization aware training schemes?
- Since the proposed scheme is a mixed-precision scheme, do you have any analysis of which layers require more bits and which layers can tolerate lower bits?
- Can you clarify when the scheme says 4-bit quantized models, it means all the layers (except input/output) use 4 or fewer bits for quantization (for both weights/activations)?
- What modifications does one need to apply this scheme to Text-to-Image diffusion models?
- Most of the experiments involve UNet style networks, do you know if the same insights would hold true for transformer networks? Similarly, currently experiments only involve DDPM/DDIM frameworks, will the similar conclusion hold true for rectified flow / EDM frameworks?
- While the performance of W6A6 scheme looks comparable to full precision scheme from the quantitative metrics on Celeba-HQ (9.01 FID vs 9.06 FID), the qualitative images in Figure 2 shows a lot of distortions in the generated images. Do you have any insights as to why this issue occurs?
- In Table 1, why does the model performance increase with lower bits (for instance, Celeba-HQ with 8/8 has better performance than full precision)?
- Can you comment on the computational cost of various methods in Table 2 ? How does the proposed scheme fare against these baselines?

Missing references:
- Stable diffusion with core ml on apple silicon : https://github.com/apple/ml-stable-diffusion
- BitsFusion: https://arxiv.org/abs/2406.04333
- LEARNED STEP SIZE QUANTIZATION https://openreview.net/pdf?id=rkgO66VKDS
- Efficientdm: Efficient quantization-aware fine-tuning of low-bit diffusion models: https://openreview.net/forum?id=UmMa3UNDAz

---

> ### Author Response · Authors · 2024-11-20
> **Response to Reviewer pc5T's Comments**
>
> We deeply thank Reviewer pc5T for the careful review and the constructive suggestions. Below please check our answers to the questions.
>
> Q1. Lack of insights into how quantization affects different model layers. Since these insights would be very helpful to understand the impact of various components in the diffusion models.
>
> Ans: Different layers react differently to quantization. Some layers are sensitive to quantization. Quantizing the sensitive layers can lead to a significant accuracy drop. For these layers, we need to allocate more bit widths to quantization. For the layers which are not sensitive to quantization, we can quantize them to smaller bit widths. We evaluated the bit allocation across layers when the model is quantized to a given size. Table 1 and Table 2 below show the results. In Table 1 and Table 2, we can see that the first layers and the last layers usually receive larger bit widths. This is because first layers are related to the input and last layers are related to the output. Layers in the middle receive smaller bit widths. We have added the discussion of the impact of quantization on different layers in the revised manuscript. The details of the bit allocation across the layers are illustrated in the newly added Figure 9 in the appendix. Thanks for this constructive suggestion.
>
> Table 1 - Bit allocation across layers when quantized to 4 bits.
> | Layer     | 10 | 20 | 30 | 40 | 50 | 60 | 70 | 80 | 90 | 100 |
> |-----------|----|----|----|----|----|----|----|----|----|-----|
> | Bit Width | 10 | 7  | 1  | 1  | 1  | 2  | 4  | 1  | 7  | 9   |
>
> Table 2 - Bit allocation across layers when quantized to 6 bits.
> | Layer     | 10 | 20 | 30 | 40 | 50 | 60 | 70 | 80 | 90 | 100 |
> |-----------|----|----|----|----|----|----|----|----|----|-----|
> | Bit Width | 10 | 9  | 3  | 1  | 2  | 4  | 8  | 6  | 9  | 10  |
>
>
>
> Q2. Since the experiments are done on smaller datasets, it would be hard to understand how the conclusions transfer to other networks / datasets / diffusion frameworks.
>
> Ans: We conducted an additional experiment on the ImageNet dataset using LDM as the framework. The visualization results have been included as Fig. 8 in the Appendix. Fig. 8 offers a visual assessment of the compressed model's performance achieved with our method on the ImageNet dataset. Given the limited time, we are not able to calculate the FID score, as we do not have the .npz file containing the distribution of the training data generated in the training stage. We need to train the model from scratch to obtain the .npz file, which is time consuming. We will add the FID score in our final manuscript after we finish the training. Thanks for this good suggestion.
>
>
> Q3. Although the quantitative metrics look good in Table 1, qualitative images in Figure 2 shows some artifacts in 6bit compression. It would be good to evaluate these more thoroughly.
>
> Ans: As we explained in the 'Evaluation Metrics' section, FID measures the distance between the distributions of generated and real images, focusing on dataset-level rather than individual image similarity. In image generation, FID calculates the distance between the distribution of the training images (ground truth) and that of the images generated from random noise. Therefore, if a generated image set has a lower FID than another, it is closer to the training set distribution, though individual images may not necessarily appear better. Additionally, images generated in full precision mode are not ground truth and thus cannot serve as a reference. Thus, some individual images generated by our compressed model may not look as good as those from the FP model, even if the overall FID is lower.
>
>
> Q4. Why use the output distortion metric (aka SSIM between output of the original and quantized models)? Since there are many other ways to define the distance between these two outputs (like mean-squared error, maximum mean discrepancy, absolute-difference, or other form of distributional distance metric)?
>
> Ans: The distance between two outputs is defined as the SSIM distance added by the Mean Squared Error (MSE) of the outputs. The SSIM distance reflects the difference in picture level, and the MSE reflects the difference in pixel level. We applied both SSIM and MSE to make the metric reflect picture-level and pixel-level similarity at the same time. In our submitted manuscript, we showed the SSIM distance in the equation, but mentioned that SSIM is added by the MSE as the final metric in the text. We have modified the equation and also included the MSE in the equation in the revised manuscript. Thanks for this good question.

---

> ### Author Response · Authors · 2024-11-20
> **Response to Reviewer pc5T's Comments - Part 2**
>
> Q5. In Eq.7, why does the total size constraint sum up both the quantized activation and the quantized weights? Shouldn't the weights and activations have separate constraint, one for the model parameters and other for the activations?
>
> Ans: We adopted the total size constraint sum up both the quantized weights and activations in the objective function. In prior works, they usually compare the results when the average size of both weights and activations is quantized to target bit widths (e.g., 6 bits or 8 bits). Similar with prior works, we adopted the total size constraint to make the comparison with them fair. It is worth mentioning that our approach can support integrating the constraints of weights and activation in a  separate way. For example, if we only quantize weights, our approach can just integrate the constraint of the size of the weights. If we quantize weights and activations to different sizes (e.g., 4-bit weights and 6-bit activations), our approach can integrate two separate constraints in the objective function.
>
>
> Q6. Have you tried only parameter quantization to see how much lower bit-widths the models can tolerate if activations are kept in higher bits (8bit or higher)?
>
> Ans: We added the results when only quantizing the weights of diffusion models. Table 3 below shows the results. As we can see in Table 3, if weights are quantized, on CIFAR-10 at DDPM, we can quantize parameters into 6 bits without hurting accuracy (loss < 0.2). Similarly, on CeleBA-HQ at DDPM, with only weights quantized, we can quantize parameters into 6 bits without hurting accuracy (loss < 0.6). We have added the results with only weights quantized in Table 4 in the appendix. Thanks for this valuable suggestion.
>
> Table 3 - Results on CIFAR-10 and CeleBA-HQ with only weights quantized.
> | Bit Width                               | CIFAT-10 (FID) | CeleBA-HQ (FID) |
> |-----------------------------------------|----------------|-----------------|
> | Full Precision                          | 3.30           | 9.01            |
> | Weights = 8 bits, Activations = 32 bits | 3.14           | 8.87            |
> | Weights = 7 bits, Activations = 32 bits | 3.15           | 8.98            |
> | Weights = 6 bits, Activations = 32 bits | 3.30           | 9.04            |
> | Weights = 5 bits, Activations = 32 bits | 6.09           | 9.27            |
> | Weights = 4 bits, Activations = 32 bits | 32.27          | 9.45            |
>
>
> Q7. How does one adapt the proposed quantization scheme for quantization aware training schemes?
>
> Ans: Our work focuses on post-training quantization, so our approach does not train or re-train the model after quantization. Note that one can seamlessly integrate our quantization scheme in the quantization-aware training. Our approach provides an optimal bit allocation with minimal output distortion for weights and activations across layers. To perform quantization-aware training, one can train the model under the optimal bit allocation provided from our approach.  During the training, one issue in the training stage is that function quantization with variable bit widths is not differentiable. To solve this issue, we can adopt Straight-Through Estimator (STE) to approximate the quantization function in back-propagation, which was discussed in [1].
>
> [1] Estimating or Propagating Gradients Through Stochastic Neurons for Conditional Computation. Yoshua Bengio, Nicholas Léonard, Aaron Courville. arXiv 2013.
>
>
> Q8. Since the proposed scheme is a mixed-precision scheme, do you have any analysis of which layers require more bits and which layers can tolerate lower bits?
>
> Ans: We analyzed the bit allocation across layers when the model is quantized to different sizes. Typically, the first layers and the last layers receive higher bit widths. However, the middle layers receive relatively lower bit widths. Please check our response to question 1 for the details. The reason is that the quantization on the first layers can affect the input values of the model and the quantization on the last layers can affect the outputs of the model. Our approach adaptively allocates the bit widths across layers to make the overall output error of the model minimized. We have added the figures of the bit allocation of the layers in Figure 9 in the appendix. Thanks for this great suggestion.
>
> Q9. Can you clarify when the scheme says 4-bit quantized models, it means all the layers (except input/output) use 4 or fewer bits for quantization (for both weights/activations)?
>
> Ans: The 4-bit quantized model means weights and activations are quantized to 4 bits on average. Because it is an average value, weights and activations in some layers may receive more than 4 bits or less than 4 bits.

---

> ### Author Response · Authors · 2024-11-20
> **Response to Reviewer pc5T's Comments - Part 3**
>
> Q10. What modifications does one need to apply this scheme to Text-to-Image diffusion models?
>
> Ans: Our approach is well generalized, which can be applied to different types of diffusion models. To apply our quantization scheme to other diffusion models (e.g., text-to-image models), one needs to extract the layers of the model and generate the rate-distortion curve for each layer to evaluate the output distortion when quantized to different bit widths. With the rate-distortion curves, we can directly apply the Lagrangian formulation proposed in our approach to solve the bit allocation problem.
>
>
> Q11. Most of the experiments involve UNet style networks, do you know if the same insights would hold true for transformer networks? Similarly, currently experiments only involve DDPM/DDIM frameworks, will the similar conclusion hold true for rectified flow / EDM frameworks?
>
> Ans: The similar conclusions can still hold true when applying our approach to other Transformer networks. Actually, the only assumption we have in our approach is that output error caused by quantization has the additivity property. We have observed the additivity property on multiple diffusion models, which has been illustrated in the submitted manuscript, and also provided the mathematical derivation for the additivity property. In any case that the additivity property holds, our approach can find the optimal bit allocation with minimized output distortion and obtain similar conclusions.
>
>
> Q12. While the performance of W6A6 scheme looks comparable to full precision scheme from the quantitative metrics on Celeba-HQ (9.01 FID vs 9.06 FID), the qualitative images in Figure 2 shows a lot of distortions in the generated images. Do you have any insights as to why this issue occurs?
>
>
> Ans: On the CelebA-HQ dataset, the FID of generated images from our compressed model is slightly lower than that from the full precision (FP) model, indicating that the distribution of our model's generated images is closer to the training set (ground truth) compared to the FP model. Two key points to note: (1) images generated by the FP model are not ground truth; (2) a lower FID score means the overall dataset is closer to the ground truth, but it does not imply that every individual image is necessarily 'better'. Therefore, it is normal to observe that some images generated by our model in Figure 2 may 'seem' less visually appealing than those generated by the FP model. This is expected, as the lower FID score indicates that, overall, the distribution of our model's generated images is closer to the training set, but it does not guarantee that every individual image will necessarily appear better. Image quality can vary, and some images may have visual imperfections despite the overall improvement in distribution.
>
>
> Q13. In Table 1, why does the model performance increase with lower bits (for instance, Celeba-HQ with 8/8 has better performance than full precision)?
>
> Ans: In the image generation domain, the ground truth for FID calculation is the training dataset, not images produced by the full precision (FP) model. Following quantization, where lower bit allocations such as 8/8 are applied, some redundant information may be reduced. This reduction can lead to slight performance improvements, as the compressed model might better capture essential features of the training distribution, sometimes even outperforming the FP model in FID despite using fewer bits.
>
> Q14. Can you comment on the computational cost of various methods in Table 2 ? How does the proposed scheme fare against these baselines?
>
> Ans: All the baseline methods in Table 2 are post-training quantization methods which do not require retraining the model after quantization. These methods are thus efficient and fast with low computational complexity. To make the comparison fair, our approach also does not retrain the model and directly compares with the baseline methods after quantization.

---

> ### Author Response · Authors · 2024-11-27
>
> Dear Reviewer pc5T,
>
> We appreciate it if you could let us know whether our responses are able to address your concerns. We are happy to address any further concerns. Thank you.
>
> Sincerely.

---

> > ### Comment · Reviewer_pc5T · 2024-12-02
> >
> > Thank you for your reply. I've updated my review to reflect my updated score. I would expect the authors to include FID comparison on ImageNet dataset.

---

> > > ### Author Response · Authors · 2024-12-03
> > >
> > > Dear Reviewer pc5T,
> > >
> > > We really appreciate your updated review and score. We will include the FID comparison in the finalized manuscript.
> > >
> > > Sincerely

---

### Author Response · Authors · 2024-11-29

Dear AC, Reviewers,

We have answered the questions from reviewers. We appreciate if reviewers can let us know whether our answers have addressed their concerns. We are glad to answer further concerns and questions. Thank you.

Sincerely,

---

### Meta-Review · Area_Chair_7e9z · 2024-12-18

**Metareview:**

This paper introduces a post-training quantization approach with mixed-precision to reduce the size and computational complexity of diffusion models. To account for varying output distributions across timesteps, the approach groups timesteps and optimizes quantization parameters for each group independently. The methods quantize models to 5–6 bits while maintaining good performance.


However, most experiments in this paper were conducted on small-scale datasets such as CIFAR-10, CelebA-HQ, and LSUN. Additionally, a reviewer raised concerns about unfair comparisons between this work and baseline approaches. The authors are encouraged to include quantitative results on large-scale datasets like ImageNet-1K and address the issues regarding comparisons.

**Additional Comments On Reviewer Discussion:**

During the authors-reviewers discussion, one reviewer participated and increased their rating. During the ACs-reviewers discussion, the reviewer with a very positive rating neither championed this paper nor responded. The reviewer with a relatively negative rating also did not provide further feedback. One reviewer expressed concerns about the comparisons conducted in this work. Overall, considering the scope of this work, the authors are encouraged to improve the quality of the submission for a future venue.

---

### Decision · Program_Chairs · 2025-01-22

Reject